# PRISM: PRIor from corpus Statistics for topic Modeling

**Tal Ishon**                                                                 *ishonta@biu.ac.il*
*Department of Computer Science*
*Bar Ilan University*

**Prof. Yoav Goldberg**                                                        *yogo@cs.biu.ac.il*
*Department of Computer Science*
*Bar Ilan University*

**Dr. Uri Shaham**                                                            *uri.shaham@biu.ac.il*
*Department of Computer Science*
*Bar Ilan University*

**Reviewed on OpenReview:** *https://openreview.net/forum?id=454v3Xbtza*

## Abstract

Topic modeling seeks to uncover latent semantic structure in text, with LDA providing a foundational probabilistic framework. While recent methods often incorporate external knowledge (e.g., pre-trained embeddings), such reliance limits applicability in emerging or underexplored domains. We introduce **PRISM**, a corpus-intrinsic method that derives a Dirichlet parameter from word co-occurrence statistics to initialize LDA without altering its generative process. Experiments on text and single cell RNA-seq data show that PRISM improves topic coherence and interpretability, rivaling models that rely on external knowledge. These results underscore the value of corpus-driven initialization for topic modeling in resource-constrained settings.

Code is available at: `https://github.com/shaham-lab/PRISM#`.

## 1 Introduction

Topic modeling is a cornerstone technique in Natural Language Processing (NLP) for uncovering latent semantic structures in text. It infers the thematic composition of a corpus by representing each topic as a probability distribution over words. The versatility of topic modeling is evidenced by its application across a spectrum of disciplines - from analyzing customer feedback in e-commerce platforms to modeling gene expression patterns in biology by treating genes as "vocabulary" and samples as "documents". Such cross-disciplinary utility underscores topic modeling's broad relevance in both applied and scientific contexts.

Since its inception, among topic modeling techniques, Latent Dirichlet Allocation (LDA) (Blei et al., 2003) remains a foundation model, leveraging Bayesian inference to estimate document-topic and topic-word distributions while inspiring numerous generative extensions. Topic modeling research has since evolved along many directions, which can be broadly categorized into two main paradigms. The first follows LDA's corpus-intrinsic approach, relying solely on statistical patterns within the target corpus through methods including graph-based and neural models that enhance semantic representation and topic coherence. The second paradigm incorporates external knowledge—such as pre-trained embeddings from large-scale language models or domain-specific priors—to guide topic discovery beyond the statistical patterns available in the input corpus alone.

Corpus-intrinsic topic modeling can be particularly attractive in emerging scientific domains where large-scale pretrained representations may not be explicitly optimized for the target modality, and domain knowledge remains incomplete or evolving. In biology, for example, key regulatory genes and functional proteins may be undiscovered or poorly characterized, limiting the reliability of external knowledge sources. While pretrained

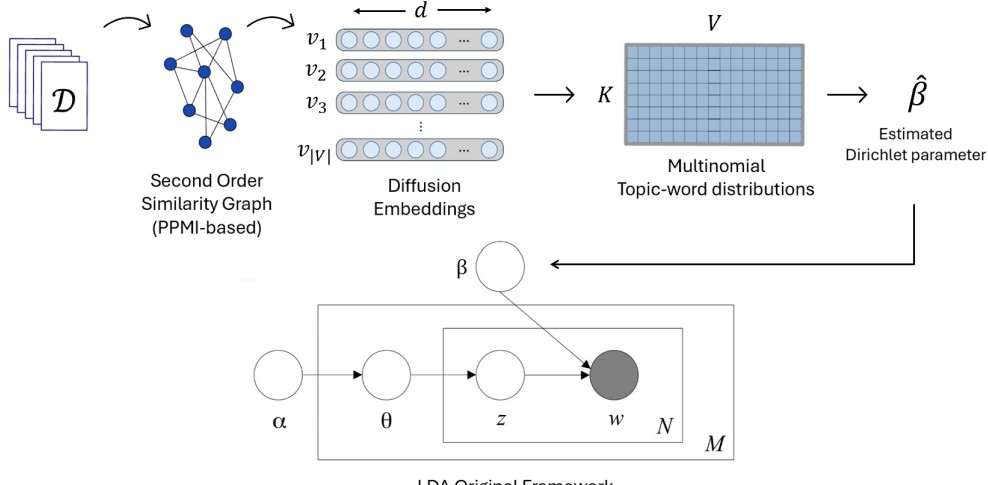

Figure 1: PRISM overview. From corpus $\mathcal{D}$, we build a second-order word-similarity graph (PPMI + cosine), obtain diffusion-map embeddings $v_i$, softly cluster them into $K$ topics, and estimate a data-driven topic–word Dirichlet prior $\hat{\beta}$ for LDA. The lower panel shows the standard LDA graphical model; PRISM replaces the symmetric $\boldsymbol{\beta}$ with $\hat{\boldsymbol{\beta}}$ while leaving the generative process unchanged.

representations are often highly effective in well-established domains such as general text, it seems prudent not to rely on them exclusively: they can encode pretraining-data artifacts and biases and introduce inductive biases that are misaligned with the target corpus (Belém et al., 2024; Ghate et al., 2025).

In this work, we introduce **PRISM**—a **PRI**or from corpus **S**tatistics for topic **M**odeling—which enhances LDA solely through corpus-intrinsic initialization (Figure 1). PRISM shapes the model's "initial perspective" by deriving informed topic-word distributions from corpus statistics, providing prior-like guidance that improves topic quality. Empirical results across five text corpora and a single-cell RNA sequencing dataset show that PRISM significantly improves topic coherence and interpretability, often matching or exceeding externally guided methods.

## 2 Related Work

Topic modeling methods broadly follow two paradigms: approaches that inject external knowledge (e.g., expert resources or pretrained representations) and corpus-intrinsic approaches that rely only on corpus statistics.

### 2.1 Methods Incorporating External Knowledge

A prominent line of work injects external semantic structure to guide topic discovery. Expert-guided models incorporate supervision such as seed words (e.g., SeededLDA (Jagarlamudi et al., 2012)) and have been extended to clinical phenotyping from EHRs (Song et al., 2022). Related approaches integrate structured resources more directly, such as medical knowledge graphs within end-to-end topic models (Zou et al., 2022).

Another route leverages pretrained representations. ETM parameterizes topic-word distributions via pretrained word embeddings (Dieng et al., 2020), while CTM amortizes inference using contextual document embeddings (Bianchi et al., 2021). Several methods induce topics directly in embedding space, including BERTopic, which clusters sentence embeddings and labels clusters via c-TF-IDF (Grootendorst, 2022), Contextual-Top2Vec, which clusters contextual token embeddings to identify dense semantic regions (Angelov & Inkpen, 2024), and FASTopic, which reconstructs semantic relations among pretrained document embeddings with learnable topic/word embeddings (Wu et al., 2024).

Overall, externally guided approaches can improve interpretability and alignment with known concepts, but they assume access to reliable curated resources or pretrained semantics; when such information is incomplete or weakly aligned–as can occur in heterogeneous, measurement-noisy biological modalities (e.g., single-cell gene expression)–these assumptions may be less stable (Kedzierska et al., 2025), motivating corpus-intrinsic alternatives.

### 2.2 Corpus-Intrinsic Methods (No External Knowledge)

Corpus-intrinsic topic models infer structure solely from the observed corpus. LDA (Blei et al., 2003) remains the canonical probabilistic formulation, while NMF (Lee & Seung, 2001) provides a non-generative factorization alternative. Neural variational topic models modernize inference under a BoW likelihood (Srivastava & Sutton, 2017), and recent work further exploits corpus-internal structure (e.g., contrastive or graph-based signals) to improve topic quality without external knowledge (Nguyen et al., 2024; Luo et al., 2024).

Complementary work has examined how Dirichlet hyperparameters influence LDA. Asuncion et al. (Asuncion et al., 2009) and Wallach et al. (Wallach et al., 2009) treat $\boldsymbol{\alpha}$ and $\boldsymbol{\beta}$ as low-dimensional concentration parameters tuned via empirical Bayes. This inquiry has been modernized in neural topic modeling, where researchers now learn these hyperparameters through backpropagation and specialized variational distributions (Burkhardt & Kramer, 2019; Joo et al., 2020). While these recent updates allow for more flexible gradients and improved global sparsity, they still primarily calibrate smoothing levels rather than imposing vocabulary-level semantic structure on the topic–word prior.

Taken together, these lines of work demonstrate the value of corpus-internal signals for enhancing topic models without external supervision. Building on this idea, PRISM introduces a corpus-derived prior as a data-driven initialization for LDA, constructing a full $V$-dimensional, semantically structured $\hat{\boldsymbol{\beta}}$ from corpus similarity. This provides a more informed starting point—a prism through which the model better captures semantic structure—while preserving LDA's generative foundations.

## 3 Preliminaries

This section introduces the key mathematical tools underpinning our approach.

### 3.1 Latent Dirichlet Allocation (LDA)

LDA (Blei et al., 2003) is a generative probabilistic model in which each document is a mixture over $K$ latent topics, and each topic is a distribution over words. For each document $d$, topic proportions $\theta_d$ are drawn from a Dirichlet distribution with parameter $\boldsymbol{\alpha}$. Each word $w_{dn}$ is then generated by first sampling a topic $z_{dn} \sim \text{Multinomial}(\theta_d)$, followed by sampling the word from the corresponding topic–word distribution, drawn from a Dirichlet distribution with parameter $\boldsymbol{\beta}$.

In the original formulation of LDA (Blei et al., 2003), $\boldsymbol{\beta}$ denotes the *multinomial topic–word distribution*. In contrast, implementations based on collapsed Gibbs sampling, including MALLET (McCallum, 2002), treat $\boldsymbol{\beta}$ as the *Dirichlet parameter* over topic–word distributions. Throughout this work we adopt the latter convention, consistent with MALLET and other Gibbs-based LDA implementations.

Inference is commonly performed using *Collapsed Gibbs Sampling* (Griffiths & Steyvers, 2004), which samples topic assignments $z_{dn}$ based on current token counts and priors - $\boldsymbol{\alpha}$ and $\boldsymbol{\beta}$. This method, implemented efficiently in MALLET (McCallum, 2002), is a widely adopted baseline.

### 3.2 Pointwise Mutual Information and Variants

Pointwise Mutual Information (PMI) (Church & Hanks, 1990) quantifies the association between two words $w_i$ and $w_j$ by comparing their joint probability to the product of their marginal probabilities. To suppress noisy or uninformative associations, Positive PMI (PPMI) retains only non-negative values.

Beyond first-order association (direct PPMI between word pairs), second-order similarity compares words via the similarity of their PPMI context profiles (i.e., cosine similarity between PPMI row vectors). Intuitively,

two words may be related even without direct co-occurrence if their contextual associations overlap; for example, *surgeon* and *physician* may seldom co-occur yet both associate with *patient*, *hospital*, and *diagnosis*. This captures semantic relatedness when words occur in similar contexts (Schütze, 1998). The use of cosine in a PPMI space is further supported by analyses linking predictive objectives such as SGNS to implicit factorization of a PPMI matrix (Levy & Goldberg, 2014). Empirically, PPMI vectors combined with cosine similarity perform competitively on semantic similarity evaluations, suggesting that corpus co-occurrence statistics already encode substantial information about word relatedness (Bullinaria & Levy, 2012).

### 3.3 Diffusion Maps

Diffusion Maps (Coifman & Lafon, 2006) provides a nonlinear dimensionality reduction technique that captures the intrinsic geometry of data represented as a similarity graph. Given an undirected word-word similarity graph $W$, the method constructs a Markov transition matrix $P$ according the following formulation: $P = D^{-1}W$, where $D$ is the diagonal degree matrix with entries $D_{ii} = \sum_j W_{ij}$. Then, it performs eigen-decomposition over $P$ to obtain the diffusion embedding. Each word is then embedded as a vector:

$$\Psi_t(w_i) = \left( \lambda_1^t \psi_1(i), \lambda_2^t \psi_2(i), \dots, \lambda_m^t \psi_m(i) \right),$$

where $\lambda_k$ and $\psi_k$ are the $k$-th eigenvalue and eigenvector of the transition matrix, respectively; $m$ is the embedding dimension, and $t$ is the diffusion time controlling the decay.

## 4 Proposed Method

We introduce **PRISM**, a corpus-intrinsic method that improves topic modeling by providing a data-driven initialization for LDA. PRISM proceeds in two stages: (1) it constructs word embeddings directly from corpus statistics by building a PPMI-based word similarity graph and applying diffusion maps to obtain low-dimensional embeddings capturing its global structure; (2) it softly clusters these embeddings to form topic-word distributions and fits a Dirichlet parameter $\hat{\boldsymbol{\beta}}$, which is used to initialize LDA. This yields prior-like guidance while leaving LDA's generative process unchanged.

### 4.1 Constructing Word Embeddings

#### 4.1.1 Similarity Graph

To capture semantic similarity between words, we construct an undirected weighted graph $W$ based on Positive PMI (PPMI), as described in Section 3.2. The PPMI matrix is computed using document-level co-occurrence, treating each document as a context window. Each word $w_i$ is represented by its corresponding row vector $\mathbf{v}_i$ from the PPMI matrix, encoding its distributional context. We then define pairwise similarity using cosine similarity as $W_{i,j} = \cos(\mathbf{v}_i, \mathbf{v}_j)$, where $\mathbf{v}_i$ and $\mathbf{v}_j$ are the PPMI vectors of words $w_i$ and $w_j$, respectively. The resulting weighted similarity graph $W$ captures both direct and indirect semantic associations by leveraging the principle that words appearing in similar contexts tend to have similar vector representations.

#### 4.1.2 From Graph to Embeddings

To obtain dense word representations from the weighted similarity graph, we apply diffusion maps, as defined in Section 3.3. This spectral embedding technique captures high-order semantic relationships by modeling multi-step transitions over the graph. Intuitively, if a diffusion process is initiated at two semantically similar words, their transition probabilities over time will be similar, reflecting shared contextual neighborhoods.

In our implementation, we construct the transition matrix $P$ from the PPMI-weighted similarity graph and apply a density-normalization step ($\alpha = 1$) following the anisotropic normalization of (Coifman & Lafon, 2006) to mitigate the influence of word frequency variations. We embed each word using the top $m$ diffusion components–specifically, the leading non-trivial right eigenvectors of $P$, scaled by their corresponding eigenvalues at diffusion time $t = 1$. We set $t = 1$ to preserve local semantic neighborhoods

and fine-grained word relationships, as higher values of $t$ would suppress high-frequency geometric detail in favor of broader, less discriminative global clusters. Empirically, selecting between 80 and 130 components yields the best semantic representation, with the optimal $m$ chosen via grid search based on topic coherence scores for each dataset.

The resulting vectors serve as our corpus-specific word embeddings, encoding semantic structure without relying on any external resources.

### 4.2 Estimating Dirichlet Parameter $\beta$

#### 4.2.1 Topic-Word Distributions

To compute empirical topic-word distributions from the learned word embeddings, we apply a Gaussian Mixture Model (GMM) (Reynolds, 2009) with $K$ components to softly cluster the word representations. The GMM yields the posterior probability $p(z \mid w)$ of each word $w$ belonging to topic $z$, along with the topic priors $p(z)$.

However, our goal is to obtain $p(w \mid z)$—the probability of a word given a topic—as required to estimate the Dirichlet parameter $\beta$ over topic-word distributions. To do so, we apply Bayes' Rule:

$$p(w \mid z) = \frac{p(z \mid w)p(w)}{p(z)},$$

where $p(z \mid w)$ is given by the GMM, $p(z)$ is the mixture weight for component $z$, and $p(w)$ is taken from unigram distribution, computed as the frequency of word $w$ in the corpus divided by the total number of tokens.

This yields multinomial topic-word distributions matrix $\boldsymbol{X} \in \mathbb{R}^{K \times V}$, where each row corresponds to $p(w \mid z)$ for a given topic. The matrix is grounded entirely in corpus-derived signals—capturing both contextual similarity and token-level frequency—used to estimate the Dirichlet parameter $\boldsymbol{\beta}$.

#### 4.2.2 Parameter $\beta$ Estimation

To estimate a Dirichlet parameter $\boldsymbol{\beta} \in \mathbb{R}^V$ over the vocabulary, we apply the method of moments (Minka, 2000), a classical statistical technique used for parameter estimation. The core idea is to match the theoretical moments of the Dirichlet distribution to empirical moments computed from data.
Let $\mathbf{X}^{(1)}, \ldots, \mathbf{X}^{(k)} \in \Delta^{V-1}$ be $k$ observed samples from a $Dirichlet(\boldsymbol{\beta}_1 \ldots \boldsymbol{\beta}_V)$ distribution over the $(V-1)-$ probability simplex. The method of moments estimator for $\hat{\boldsymbol{\beta}}$ is given by

$$\hat{\boldsymbol{\beta}}_i = \mathbb{E}[X_i] \left( \frac{\mathbb{E}[X_j](1 - \mathbb{E}[X_j])}{\mathbb{V}[X_j]} - 1 \right),$$

where, following Minka (2000), we set $i = j$, and

$$\mathbb{E}[X_i] \approx \frac{1}{k} \sum_{\ell=1}^{k} X_i^{(\ell)}, \qquad \mathbb{V}[X_i] \approx \frac{1}{k-1} \sum_{\ell=1}^{k} \left( X_i^{(\ell)} - \mathbb{E}[X_i] \right)^2.$$

#### 4.2.3 Initializing LDA with $\hat{\beta}$

The estimated vector $\hat{\boldsymbol{\beta}}$ is then used to initialize the LDA model, replacing the standard uniform or fixed scalar prior. We modified the MALLET implementation to support a vector valued $\hat{\boldsymbol{\beta}}$ parameter, enabling topic-word distributions to reflect corpus-specific semantic structure from the outset. Apart from this change, the rest of the LDA inference pipeline in MALLET remains unaltered.

## 5 Experiments

We evaluate the effectiveness of our corpus-informed Dirichlet parameter by assessing its impact on standard LDA. Our goal is to determine whether data-driven initialization can substantially improve topic quality and

bring LDA closer to, or even surpass, state-of-the-art topic modeling methods. Experiments are conducted on five diverse text corpora, using three complementary metrics. (Ablation studies appear in Appendix D.2, and the complexity analysis in Appendix D.4).

**Datasets.** To ensure a fair and objective evaluation, we use pre-processed datasets from the OCTIS framework (Terragni et al., 2021), thereby avoiding model-specific tuning of the preprocessing pipeline. Specifically, we experiment with four diverse OCTIS datasets: 20NewsGroup, BBC News, M10, and DBLP, covering a range of domains and document styles.

In addition, following BERTopic (Grootendorst, 2022), we include the TrumpTweets (TT) dataset to test model performance on informal, short-form text. Since this dataset is not included in OCTIS, we apply OCTIS's preprocessing module with standard, commonly used filtering (see Appendix A for details). This setup allows us to evaluate our method both under standardized preprocessing and in a setting closer to real-world social media text. The statistical details can be seen in Appendix A.

**Baselines.** We evaluate our approach against a diverse set of topic models. For classical and neural baselines implemented in OCTIS (Terrone et al., 2021), we include LDA (Blei et al., 2003), NMF (Zhao et al., 2017), ProdLDA (Srivastava & Sutton, 2017), NeuralLDA (Srivastava & Sutton, 2017), and the ETM (Dieng et al., 2020). These models rely solely on corpus-internal signals and do not incorporate any external knowledge.

We further compare against recent embedding-based methods that leverage pretrained representations: BERTopic (Grootendorst, 2022), FASTopic (Wu et al., 2024), and Contextual Contextual-Top2Vec (Angelov & Inkpen, 2024). These models utilize external knowledge, typically through pretrained sentence embeddings such as MiniLM or the Universal Sentence Encoder. We follow each method's official implementation and recommended configuration as provided in their respective GitHub repositories.

Additionally, we include MALLET (McCallum, 2002), a highly optimized Gibbs-sampling-based LDA implementation, and our proposed model, PRISM. Both were run using the MALLET framework with default hyperparameters, enabling internal parameter optimization (e.g., `optimizeInterval`). For PRISM, we further supplied the model with estimated $\hat{\boldsymbol{\beta}}$ as described in Section 4.

**Metrics.** We evaluate topic models using statistical and human-aligned metrics that capture different dimensions of quality: $c_v$ *Coherence*, normalized pointwise mutual information (NPMI) and the word intrusion detection (*WID*) task. Formal definitions appear in Appendix B. $c_v$ *Coherence* (Röder et al., 2015) and NPMI (Bouma, 2009) measure the semantic relatedness of top words within a topic, based on co-occurrence statistics. While both are widely used, we tend to favor $c_v$ due to its empirically stronger correlation with human judgments. *WID* (Chang et al., 2009) evaluates topic interpretability via the identification of an out-of-place word among a topic's top words. To scale this human-centric task, we follow (Garg et al., 2023) and use large language models (LLMs) as automated judges. WID implementation details are provided in Appendix B.2.

**Setup.** To enable fair and robust comparison, we evaluate all models under a unified protocol. For each dataset, we select three values of $K$ near the reference number of ground-truth topics, along with slightly larger values to support finer-grained topic discovery. For TrumpTweets, which lacks labels, we use comparable $K$ values to those in labeled datasets (See Table 4). This protocol reflects a weakly-supervised topic modeling paradigm, in which coarse supervision guides model behavior without enforcing strict topic-label alignment, following the principles outlined by (Zhao et al., 2017). All models are evaluated using the top 10 words per topic, and each configuration is run 10 times to account for stochastic variation; final scores are averaged across runs and topic counts.

Models with adaptive topic selection are evaluated accordingly. For BERTopic, we report the best result across runs with and without a fixed $K$; for Contextual-Top2Vec, which infers $K$ automatically, we evaluate its output as-is. Further implementation details appear in Appendix C.

Table 1: Model performance across datasets. Each entry is mean (std) over three topic settings. Best is **bold**, second-best is underlined. † indicates PRISM outperforms MALLET for that metric.

| MODELS | 20NG | | BBC News | | M10 | | DBLP | | TrumpTweets | |
|---|---|---|---|---|---|---|---|---|---|---|
| | $c_v$ | NPMI | $c_v$ | NPMI | $c_v$ | NPMI | $c_v$ | NPMI | $c_v$ | NPMI |
| ProdLDA | .5976 (.0216) | .0569 (.0057) | .6427 (.0059) | −.0046 (.0110) | .4218 (.0336) | −.0951 (.0954) | **.4733** (.0196) | .0072 (.0464) | .5373 (.0099) | .0758 (.0029) |
| NeuralLDA | .5339 (.0049) | .0425 (.0010) | .5720 (.0234) | −.0532 (.0274) | .4179 (.0183) | −.1950 (.0459) | .3833 (.1022) | −.0207 (.0768) | .4252 (.0159) | −.0249 (.0037) |
| LDA | .5321 (.0078) | .0457 (.0033) | .4924 (.0150) | −.0409 (.0153) | .3833 (.0116) | −.1602 (.0455) | .3647 (.0148) | −.0273 (.0262) | .4188 (.0196) | −.0109 (.0041) |
| ETM | .5242 (.0079) | .0433 (.0049) | .4963 (.0062) | −.0176 (.0103) | .3659 (.0150) | −.1255 (.0384) | .2828 (.0581) | −.0389 (.0152) | .4133 (.0338) | .0203 (.0136) |
| NMF | .5497 (.0087) | .0550 (.0029) | .4564 (.0369) | −.0037 (.0110) | .3536 (.0019) | −.0943 (.0487) | .3663 (.0213) | .0102 (.0188) | .4358 (.0103) | .0326 (.0120) |
| BERTopic | .5557 (.0155) | .0887 (.0113) | **.7124** (.0073) | **.1694** (.0022) | .4030 (.0131) | **.1310** (.0095) | .3862 (.0075) | −.0039 (.0051) | .4461 (.0025) | −.0287 (.0043) |
| C-Top2Vec | .5632 (.0327) | .0682 (.0152) | .5108 (.0484) | −.5750 (.0262) | .3543 (.0033) | −.1675 (.0036) | .3185 (.0005) | −.1328 (.0019) | .3653 (.0028) | −.2349 (.0045) |
| FASTopic | .5744 (.0258) | .0223 (.0078) | .6572 (.0314) | .0236 (.0136) | .4473 (.0081) | −.2065 (.0172) | .3239 (.0027) | .0012 (.0102) | .3804 (.0243) | −.1319 (.0380) |
| MALLET | .6322 (.0032) | .1018 (.0037) | .6384 (.0051) | .1285 (.0101) | .4571 (.0058) | .0671 (.0032) | .4329 (.0133) | .0467 (.0083) | .4989 (.0152) | .0701 (.0049) |
| **PRISM (ours)** | **.6592**†(.0081) | **.1168**†(.0061) | .6781†(.0161) | .1468†(.0179) | **.5285**†(.0034) | .0822†(.0132) | .4643†(.0152) | **.0751**†(.0131) | **.5571**†(.0036) | **.0991**†(.0112) |

Table 2: Word Intrusion Detection accuracy across five datasets (mean (std) over three topic settings). Best is **bold**, second-best is underlined; † indicates PRISM outperforms MALLET.

| MODELS | 20NewsGroup | BBC | M10 | DBLP | TrumpTweets |
|---|---|---|---|---|---|
| ProdLDA | .4561 (.0165) | .4556 (.0681) | .2232 (.0322) | .2528 (.0638) | .2759 (.0722) |
| NeuralLDA | .2533 (.0518) | .3259 (.0060) | .1194 (.0211) | .1667 (.0667) | .1296 (.0414) |
| LDA | .1472 (.0202) | .1333 (.3615) | .1139 (.4885) | .0833 (.9211) | .1611 (.5625) |
| ETM | .3422 (.0301) | .4592 (.0726) | .0889 (.3713) | .1194 (.6224) | .1130 (.9177) |
| NMF | .1411 (.0196) | .0630 (.3148) | .0010 (.8161) | .0010 (.8161) | .2426 (.3977) |
| BERTopic | .3811 (.0452) | .5537 (.0092) | **.4907** (.0227) | .3584 (.0118) | .3139 (.0103) |
| C-Top2Vec | .6022 (.0309) | .5915 (.0618) | .3875 (.0188) | **.4928** (.0356) | **.3554** (.1016) |
| FASTopic | .5844 (.0265) | **.6315** (.0500) | .3379 (.0442) | .2722 (.0278) | .2352 (.0466) |
| MALLET | .5681 (.0206) | .4689 (.0412) | .3711 (.0406) | .3031 (.0554) | .2697 (.0302) |
| **PRISM (ours)** | **.6099**†(.0124) | .6201†(.0281) | .3921†(.0176) | †.3408 (.0512) | †.3057 (.0131) |

**Our Results.** We evaluate PRISM across five benchmark datasets, focusing on two key questions: (1) whether it provides consistent improvements over classical LDA as implemented in MALLET, and (2) how it compares to recent topic models, including both corpus-intrinsic approaches and those that leverage external semantic knowledge. The following results address both aspects. Models above the dashed line are direct baselines; those below use external embeddings and represent an upper bound (Table 1, Table 2).

**Quantitative Results.** As shown in Table 1, PRISM consistently outperforms the original MALLET implementation, achieving substantial gains in both $c_v$ and NPMI. Beyond improving upon MALLET, PRISM frequently closes the gap to, or even surpasses, recent embedding-based methods. This is particularly evident on the BBC News, M10, and TrumpTweets datasets, where PRISM not only significantly outperforms MALLET but also achieves the best or second-best scores across both metrics—demonstrating competitiveness with SOTA methods. PRISM obtains the best $c_v$ score on three out of five datasets (20NG, M10, TrumpTweets), and ranks second on BBC News and DBLP. Given that $c_v$ is widely regarded as more reflective of human topic judgments, these results suggest that PRISM produces more interpretable and semantically coherent topics across diverse domains. While NPMI improvements are somewhat more modest, PRISM still achieves the best scores on 20NewsGroup, M10 and TrumpTweets and remains competitive throughout.

To further assess topic interpretability, we employ the Word Intrusion Detection (WID) task (detailed in Appendix B.2). As shown in Table 2, PRISM ranks among the top two models on three out of five datasets and remains highly competitive on the remaining two. Among corpus-intrinsic models (above the dashed line), PRISM consistently achieves the highest accuracy—outperforming MALLET and all other classical baselines. It also competes strongly with embedding-based methods (below the dashed line), outperforming all of them on 20NG and several on BBC and M1. Overall, these results show that PRISM not only dominates traditional models in both coherence and interpretability but also matches—and at times exceeds—the performance of state-of-the-art models that rely on external knowledge.

**Qualitative Analysis.** Over BBC dataset, we show a comparison of PRISM to BERTopic, which achieved the highest $c_v$ and NPMI scores, and to ProdLDA, the strongest corpus-intrinsic baseline in $c_v$ (Table 1)

**(a) BERTopic: top-10 words**

| Topic 1 | Topic 2 | Topic 3 | Topic 4 | Topic 5 |
|---|---|---|---|---|
| government | film | win | game | virus |
| election | good | play | phone | mail |
| party | award | player | mobile | site |
| labour | music | match | technology | software |
| plan | win | game | device | program |
| tory | show | final | service | security |
| rise | star | club | music | user |
| country | include | good | video | attack |
| tax | actor | back | search | computer |
| economy | band | team | gadget | net |

**(b) ProdLDA: top-10 words**

| Topic 1 | Topic 2 | Topic 3 | Topic 4 | Topic 5 |
|---|---|---|---|---|
| mobile | category | bug | rise | election |
| game | victory | yesterday | quarter | government |
| phone | aviator | shrink | interest | plan |
| player | album | navigate | investment | party |
| technology | film | keynote | rate | labour |
| firm | goal | fate | growth | issue |
| service | performance | declaration | analyst | tory |
| user | great | finnish | figure | leader |
| music | ball | excitement | price | conservative |
| system | band | inevitable | stock | public |

**(c) PRISM: top-10 words**

| Topic 1 | Topic 2 | Topic 3 | Topic 4 | Topic 5 |
|---|---|---|---|---|
| government | film | company | win | technology |
| election | good | market | game | computer |
| party | award | firm | play | phone |
| labour | music | rise | player | mobile |
| plan | win | sale | good | service |
| tory | show | price | back | user |
| law | include | economy | match | game |
| issue | star | share | team | firm |
| public | top | growth | club | net |
| minister | actor | month | final | music |

Figure 2: Top-10 words per topic on the **BBC** dataset with $K = 5$. Each column is a distinct topic. Colors denote manually interpreted categories (politics, entertainment, business, sports, technology); lighter shades indicate weaker relevance and white indicates no clear association. Panels: (a) BERTopic, (b) ProdLDA, (c) PRISM.

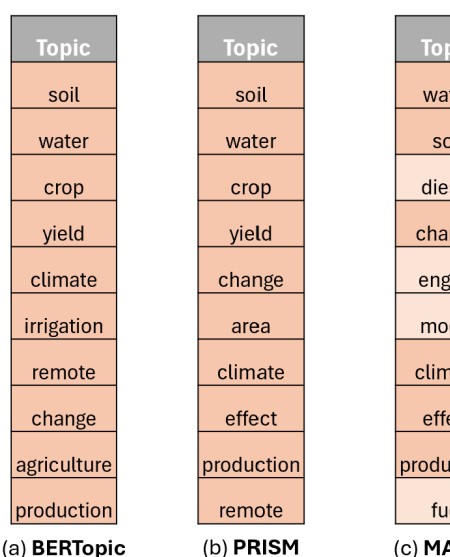

**Figure 3**

| (a) BERTopic | (b) PRISM | | (c) MALLET | |
|---|---|---|---|---|
| Topic | Topic | Topic | Topic | Topic |
| gene | gene | protein | gene | protein |
| expression | expression | structure | expression | sequence |
| protein | datum | sequence | datum | model |
| datum | microarray | dna | model | structure |
| sequence | analysis | model | network | dna |
| cluster | cluster | motif | cluster | motif |
| dna | regulatory | fold | analysis | base |
| microarray | model | prediction | microarray | fold |
| model | method | algorithm | regulatory | prediction |
| fold | classification | bind | paper | bind |

Figure 3: Top-10 words for biology topics in M10.

**Figure 4**

| (a) BERTopic | (b) PRISM | (c) MALLET |
|---|---|---|
| Topic | Topic | Topic |
| soil | soil | water |
| water | water | soil |
| crop | crop | diesel |
| yield | yield | change |
| climate | change | engine |
| irrigation | area | model |
| remote | climate | climate |
| change | effect | effect |
| agriculture | production | production |
| production | remote | fuel |

Figure 4: Top-10 words for a climate/agriculture topic in M10.

over BBC. The dataset contains five ground-truth topic labels. As shown in Figure 2, PRISM successfully recovers all five topics—politics, business, sports, technology, and entertainment—with minimal off-topic words. In contrast, BERTopic fails to recover the business category and exhibits redundancy across two overlapping technology topics. ProdLDA clearly captures politics and business, while entertainment and technology are only partially distinguishable, and sports is entirely missing. These observations align with the WID results (Table 2), where PRISM ranks second overall, while BERTopic and ProdLDA rank fourth and seventh, respectively. This suggests that PRISM produces more coherent and uniquely identifiable topics. Supplementary examples and heatmap comparisons are included in Appendix D.1.

On M10, we compare PRISM to BERTopic and MALLET. PRISM achieves the highest $c_v$ and second-best WID score, while BERTopic leads in WID and NPMI, and MALLET ranks second in $c_v$ (Table 1, Table 2). As shown in Figure 3, all models capture biologically meaningful themes, though MALLET includes a few less relevant terms (e.g., "paper," "network"). BERTopic merges gene-related and protein-related concepts into a single broad topic, whereas both PRISM and MALLET separate them into two distinct but related topics, one focused on gene expression and microarray, the other on protein structure and binding. This finer-grained separation reflects PRISM's stronger topical coherence and is also evident in its ranking of more meaningful terms (e.g., "microarray," "regulatory," "structure") with higher probability than MALLET. Interestingly, BERTopic's broader topic structure may contribute to its higher WID score: in the WID

task, intruder words are sampled from other high-probability topics (details in Appendix B.2), and when topics overlap semantically, as with genes and proteins, intruders may feel topically adjacent rather than clearly out of place. PRISM's more specific topics make intruder detection more challenging, which may explain its slightly lower WID despite better topic distinctiveness. Figure 4 presents another M10 topic, likely related to climate and agriculture. PRISM and BERTopic show high overlap in top-ranked words ("soil," "water," "crop," "yield," etc.), with strong topic focus. In contrast, MALLET's output contains generic or loosely related terms ("engine," "fuel," "model"), leading to reduced coherence. These results support the quantitative findings, where PRISM outperforms MALLET across all metrics and approaches the interpretability of BERTopic without using external knowledge.

## 6 Biological Experiments

**Motivation and Analogy.** We investigate the applicability of PRISM to biological data, aiming to uncover latent Biological Processes (BPs) from single-cell RNA sequencing (scRNA-seq) data. This task naturally parallels topic modeling: cells correspond to documents, genes to words, and BPs to topics. As in text, where documents often span multiple topics and words can take on different meanings depending on context, each cell may be involved in multiple BPs, and individual genes may participate in several biological functions, reflecting the many-to-many relationships captured by topic models. Furthermore, scRNA-seq data is organized as a count matrix, where each entry denotes the expression level of a gene in a cell, directly analogous to the word-document count matrix in LDA.

**Datasets.** We evaluate our approach on three scRNA-seq datasets spanning human and mouse tissues. BreastCancer comprises human breast tumor samples provided by a collaborating laboratory in a preprocessed form,[1] PBMC3k contains peripheral blood mononuclear cells (PBMCs) from a healthy donor,[2] and Zeisel brain profiles cells from the mouse cerebral cortex.[3]. All datasets were preprocessed using a standard Scanpy workflow; no provider-specific preprocessing was applied. Details are given in Appendix E.1).

**Baselines.** As a proof of concept, we compare PRISM to the standard LDA implementation in MALLET to test whether our corpus-intrinsic initialization improves the interpretability of biological processes in scRNA-seq data. We also include two widely used text-topic baselines: ProdLDA (Srivastava & Sutton, 2017) and NMF (Lee & Seung, 2001). Finally, we benchmark against two single-cell–specific factorization models: scHPF (single-cell hierarchical Poisson factorization) (Levitin et al., 2019) and cNMF (consensus NMF) (Kotliar et al., 2019).

**Evaluation Metrics.** We assess biological plausibility with an LLM-based scoring procedure following Hu et al. (2025). For each topic, we submit its top 20 genes to GPT-4 using a fixed prompt asking how likely the genes are to co-participate in a Biological Process (BP), and record the returned confidence as the topic's score. Averaging over topics yields a proxy for biological coherence (details in Appendix E.2.2). In addition, we report conventional gene-set quality metrics; definitions and full results appear in Appendix E.2.1 (definitions) and Appendix E.3 (results).

**Experimental Setup.** We evaluate all baselines and PRISM on the same scRNA-seq datasets under a matched configuration: the number of topics is fixed to $K = 5$, and each model is trained for 10 independent runs per dataset. For PRISM, the $\beta$ prior is estimated exactly as in our text-corpus experiments, with a minor adjustment to the PPMI graph construction tailored to gene–gene co-occurrence (see Appendix E.1 for details).

---

[1] https://zenodo.org/records/10620607
[2] https://scanpy.readthedocs.io/en/stable/generated/scanpy.datasets.pbmc3k.html
[3] https://www.ncbi.nlm.nih.gov/geo/query/acc.cgi?acc=GSE60361

Table 3: Gene-set quality metrics for biological datasets. Each entry is the mean over 10 runs with $K = 5$. Coherence = mean within-set Spearman correlation; Coverage = fraction of pathway members recovered (averaged over significant pathways); Strength = $-\log_{10}(q)$ (FDR-adjusted). "—" indicates no significant pathway enrichments ($q \geq 0.05$). Best scores are **bold**; second-best are underlined; † indicates PRISM significantly outperforms MALLET for that metric (test details in the Supplement).

| Models | BreastCancer | | | PBMC3k | | | Zeisel brain | | |
|---|---|---|---|---|---|---|---|---|---|
| | Coherence | Coverage | Strength | Coherence | Coverage | Strength | Coherence | Coverage | Strength |
| cNMF | 0.2695 | 0.0 | — | **0.3867** | **0.8** | 10.2447 | **0.6306** | **0.8** | **4.1679** |
| scHPF | 0.2423 | **0.2** | **1.4056** | 0.1694 | 0.6 | 6.0057 | 0.4049 | 0.6 | 3.4919 |
| ProdLDA | 0.0956 | 0.0 | — | 0.0091 | 0.0 | — | 0.0273 | 0.0 | — |
| NMF | 0.0918 | 0.0 | — | 0.0068 | 0.0 | — | 0.0395 | 0.0 | — |
| MALLET | 0.2471 | 0.0 | — | 0.2952 | **0.8** | 9.0515 | 0.5541 | **0.8** | 2.2903 |
| PRISM | **0.2906**† | 0.0 | — | 0.3488† | **0.8** | **11.8848**† | 0.5987† | **0.8** | 3.7403† |

**Results.** As shown in Table 3, two factors largely explain the observed patterns. **(i) Dataset scale.** The BreastCancer matrix contains only 297 genes (vs. 5,000 for PBMC3k/Zeisel), which limits the pathway universe and reduces statistical power for enrichment testing; accordingly, most methods show no significant enrichments on BreastCancer despite reasonable within-set coherence. **(ii) Model suitability.** Methods geared toward text without biological inductive bias (ProdLDA, NMF) perform poorly across datasets–exhibiting low coherence and no enrichment–whereas corpus-intrinsic, count-based topic models (MALLET, PRISM) and single-cell specific baselines (scHPF, cNMF) recover biologically meaningful structure. Across datasets, PRISM is consistently competitive with - and often exceeds - scRNA-seq baselines, while also outperforming MALLET on every dataset, yielding higher coherence and stronger enrichments wherever pathways are detectable. Overall, these results may indicate that LDA (as implemented in MALLET) is a strong choice for biological gene-program discovery, and that domain-appropriate, corpus-intrinsic initialization, as in PRISM, further improves the biological interpretability of the inferred programs. Another LLM-based evaluation can be found in Appendix E.3.

## 7 Conclusion

We introduced PRISM, a corpus-driven initialization method for LDA that integrates semantic structure derived directly from the data, without relying on external embeddings. Across five diverse datasets—spanning news, social media, and biomedical texts—PRISM consistently improves coherence and interpretability over classical baselines and rivals embedding-based models. Its strong performance, despite operating entirely on corpus-internal signals, highlights the underexplored potential of structure-aware initialization in probabilistic models. This work opens new directions for enhancing topic models through data-intrinsic semantics—bridging the gap between classical transparency and modern representational strength.

### 7.1 Limitations and Future Directions

PRISM, like classical topic models, requires a priori selection of $K$, treated as a tunable hyperparameter; while this may limit full automation across heterogeneous corpora, our experiments indicate robustness across a practical range of $K$. A natural extension is a corpus-driven initialization for the document–topic prior $\boldsymbol{\alpha}$, aiming to adapt sparsity and enhance interpretability, and potentially further stabilizing training across datasets.

## Acknowledgments

This research was funded in part by MOST grant number 8491/25.

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

## A  Datasets

As described in Sec 5, we evaluate on five benchmarks. Four are standard corpora from `OCTIS` (preprocessed by its internal pipeline). We also include *TrumpTweets*, following BERTopic (Grootendorst, 2022), to assess performance on short, noisy social media text. Summary statistics are in Table 4.

Table 4: Dataset statistics. Avg. Len. denotes average document length in tokens; $K$ lists the topic counts used.

| Dataset | #Docs | Vocab | Avg. Len. | #Labels | $K$ |
|---|---|---|---|---|---|
| 20NewsGroup | 16,309 | 1,612 | 48.0 | 20 | 20, 25, 50 |
| BBC | 2,225 | 2,949 | 12.1 | 5 | 5, 10, 15 |
| M10 | 8,355 | 1,696 | 5.9 | 10 | 10, 15, 20 |
| DBLP | 54,595 | 1,513 | 5.4 | 4 | 4, 10, 15 |
| TrumpTweets | 18,239 | 1,988 | 9.0 | – | 10, 15, 20 |

The *TrumpTweets* dataset was obtained from the same source cited by BERTopic.[4] To ensure consistency across datasets, we applied the `OCTIS` preprocessing module with basic filtering:

```
Preprocessing(
    vocabulary=None,
    lowercase=True,
    remove_numbers=True,
    min_words_docs=3,
    min_chars=3,
    min_df=.01,
    max_df=.9,
    max_features=2000,
    remove_punctuation=True,
    lemmatize=True,
    stopword_list="english"
)
```

This configuration balances document retention and vocabulary quality, which is especially important for short texts such as tweets.

## B  Metrics

We assess topic quality using corpus-intrinsic coherence ($c_v$) and normalized pointwise mutual information (NPMI), plus a language-model–assisted interpretability evaluation.

### B.1  Standard Topic Modeling Metrics

$c_v$ **Coherence.**  The $c_v$ metric (Röder et al., 2015) combines pairwise NPMI with cosine similarity over context vectors derived from a sliding window:

$$C_v = \frac{1}{|W|(|W|-1)} \sum_{i<j} \mathrm{NPMI}(w_i, w_j) \cdot \cos(\mathbf{v}_i, \mathbf{v}_j), \tag{1}$$

where $W$ is the set of top-$N$ words per topic and $\mathbf{v}_i \in \mathbb{R}^{|C|}$ encodes co-occurrence statistics of $w_i$ across the corpus $C$. We compute $c_v$ with `Gensim`'s `CoherenceModel` (defaults).

---

[4]`https://www.thetrumparchive.com/faq`

**NPMI.** Following (Bouma, 2009),

$$\text{NPMI}(w_i, w_j) = \frac{\log \frac{P(w_i, w_j)}{P(w_i)P(w_j)}}{-\log P(w_i, w_j)}, \tag{2}$$

which normalizes PMI to $[-1, 1]$ for comparability across corpora.

### B.2 Word Intrusion Detection (WID)

To complement statistical metrics with a proxy for human interpretability, we use the Word Intrusion Detection (WID) task. General framework of the metric can be viewed in Figure 5.

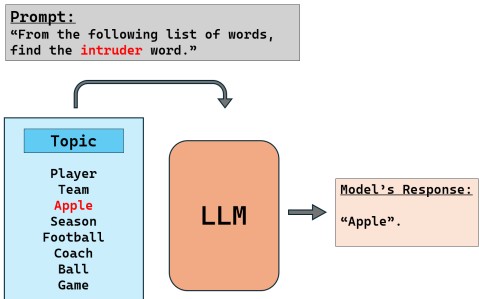

Figure 5: Illustration of the Word Intrusion Detection (WID) framework. A large language model is prompted to identify the word that does not belong in a list of top topic words (a.k.a. the intruder). The prompt shown here is illustrative; actual prompts used in our experiments follow a more structured format.

The Word Intrusion Detection (WID) task (Chang et al., 2009) is a widely adopted human-centered evaluation method for assessing topic interpretability. In this task, annotators are shown the top-$N$ words of a topic, one of which is an intruder—i.e., a word drawn from another topic that appears with low probability in the target topic but is prominent elsewhere. The annotator is asked to identify the word that does not semantically belong. Higher topic coherence typically results in easier and more consistent intruder identification, making WID an indirect yet effective proxy for human interpretability.

Recent studies have proposed leveraging large language models (LLMs) to automate WID (Garg et al., 2023), enabling scalable, consistent, and human-aligned evaluation. We adopt this paradigm by employing a LLM as an automatic evaluator within our WID framework (see Figure 5). This model is prompted to identify the intruder from each modified topic word list, effectively simulating human judgment without the need for manual annotation.

**Pipline.** Our pipeline leverages HuggingFace's `transformers` library. We initialize the tokenizer via `AutoTokenizer.from_pretrained`, explicitly setting the end-of-sequence (`eos`) token as the padding token to ensure consistent handling of short text inputs. The LLM is integrated with the `pipeline` API using `device_map="auto"` for efficient hardware mapping and `torch_dtype=torch.bfloat16` to reduce memory overhead.

During inference, each topic's word list is modified by injecting one intruder word. The model is then prompted to identify the semantic outlier. Its success in this task reflects the semantic cohesion of the topic, thus serving as an indirect interpretability metric that complements statistical scores.

The task involves identifying an intruder word inserted into an otherwise coherent topic word list. We evaluate performance using the Meta-LLaMA-3.3-70B-Instruct model (AI, 2024), which demonstrates strong alignment with human judgment.

```
"You are an assistant in a word intrusion detection task.

From the following list of tokens, identify the one token that does not belong
with the others.
For example: for 1. Banana 2. Orange 3. Japan 4. Strawberry 5. Tree The expected
answer is index 3.

The reason 3 is the intruder's index is since Banana, Orange, Strawberry and
Tree are related to fruits in a way, but Japan is a country.

Another example: for 1. Dog 2. Cat 3. Horse 4. Apple 5. Pig The expected answer
is index 4.

The reason 4 is the intruder's index is since Dog, Cat, Horse and Pig are
different kinds of animals, while Apple is a fruit.

Here are your words:  {numbered_word_list}.

In your response, you have to provide only one index between 1 to {len(words_list)},
where the index is the intruder word's index from the given list. Without any
additional explanations."
```

Figure 6: Prompt template for the Word Intrusion Detection (WID) task, with instructions, two illustrative examples, and a structured response format. The input variable `numbered_word_list` is inserted at evaluation time to test different word sets.

---

**Algorithm 1** Word Intrusion Detection (WID) Pipeline

---

**Input:** set of topic models $\mathcal{M}$; prompt template
**Output:** saved LLM outputs; model-wise accuracy scores

**for each** model $M \in \mathcal{M}$ **do**
    **for each** topic $T \in M.\text{TOPICS}$ **do**
        **for each** word list size $s \in \{10, 15, 20\}$ **do**
            $W \leftarrow \text{TOPWORDS}(T, s)$
            $W_{\text{idx}} \leftarrow \text{ADDINDICES}(W)$
            $prompt \leftarrow \text{FORMATPROMPT}(W_{\text{idx}})$
            $result \leftarrow \text{QUERYLLM}(prompt)$
            $\text{SAVEOUTPUT}(result)$

$\text{EVALUATEACCURACY}(\mathcal{M})$                                       ▷ Eq. 3

---

**Prompt Engineering.** Inspired by chain-of-thought prompting (Wei et al., 2022) and role-play prompting (Kong et al., 2023), we crafted prompts to guide the large language model (LLM). The template includes two examples that demonstrate both intruder identification and the expected output format (Figure 6).

**Evaluation Metric.** Accuracy is computed as the proportion of correct intruder identifications:

$$\text{Accuracy} = \frac{\text{Count}(\text{LLM response} = \text{true intruder})}{K}, \tag{3}$$

where $K$ is the number of topics. We report accuracy separately for top-10, top-15, and top-20 word lists.

This pipeline evaluates topic coherence by detecting intruder words—terms that do not fit within topic word lists—using an LLM. For each model, we form top-10/15/20 word lists per topic, index the words (ADDINDICES), format a structured prompt (FORMATPROMPT), query the LLM (QUERYLLM), and save outputs. We then compare predictions to the known intruders to compute accuracy (EVALUATEACCURACY).

## C  Setup

### C.1  Models

To ensure fair and reproducible comparisons, we evaluate a diverse set of topic modeling baselines using publicly available implementations under their recommended configurations.

**OCTIS Models.**  We run `LDA`, `NMF`, `ETM`, `ProdLDA`, and `NeuralLDA` via the `OCTIS` framework, using its standardized preprocessing pipeline and default hyperparameters unless noted.

**ETM configurations.**  We consider two ETM variants:

- **Without pretrained embeddings:** the default `OCTIS` configuration.

- **With pretrained embeddings:** GloVe initialization:

  ```
  ETM(num_topics=TOPICS_NUM,
      embeddings_path="filtered_glove.100d.vec",
      embedding_size=100)
  ```

Results were similar across the two settings (no detectable improvement from GloVe), so we report the corpus-only variant in the main tables.

**BERTopic.**  We use the official implementation[5] with default parameters, except we explicitly set the number of topics to match our experimental setup; when applicable, we report the better of auto-detected vs. fixed-$K$ configurations.

**Contextual Top2Vec.**  We run C-Top2Vec in contextual-embedding mode with:

```
Top2Vec(
    documents,
    split_documents=True,
    contextual_top2vec=True,
    embedding_model="all-MiniLM-L6-v2",
    speed="deep-learn",
    workers=2
)
```

This follows the recommended usage from the official repository.[6] We also tested a custom tokenizer, which did not improve performance; therefore, we use default tokenization in reported results.

**FASTopic.**  We use the `TopMost` implementation[7] with the recommended preprocessing utility:

```
preprocessing = Preprocess(stopwords="English")
model = FASTopic(num_topics=topic_num, preprocess=preprocessing)
```

Hyperparameters follow the defaults in the official GitHub repository,[8] with no additional tuning.

### C.2  Computing Infrastructure

**Local (WSL).** ASUS Vivobook S 15/16 OLED; Windows 11 + WSL2 (Ubuntu). CPU: *Intel Core Ultra 9 185H* (11 cores / 22 threads); RAM: *16 GB*. GPU: not utilized. Used for PRISM, MALLET, and OCTIS models (LDA, NMF, ProdLDA, ETM, NeuralLDA), and BERTᴏᴘɪᴄ.

---

[5]https://github.com/MaartenGr/BERTopic
[6]https://github.com/ddangelov/Top2Vec
[7]https://github.com/yfsong0709/TopMost
[8]https://github.com/bobxwu/FASTopic

**Server.** Linux (Ubuntu). GPUs: *4× NVIDIA L40S*, each with ∼46 GiB VRAM (driver *575.57.08*, CUDA *12.9*). Example utilization snapshot (from `nvidiasmi`): GPU 0/1/3 active (≈28.6/17.9/17.1 GiB). (Host CPU/RAM not recorded.) Used for FASTopic and CONTEXTUAL-TOP2VEC.

## C.3 Licenses of External Assets

We list the license for each third-party asset used.

Table 5: Licenses / terms for third-party software and assets.

| Asset | License / Terms |
|---|---|
| *Software / libraries* | |
| OCTIS | MIT License[9] |
| Gensim | LGPL v2.1 (or later)[10] |
| MALLET | Apache License 2.0[11] |
| BERTopic | MIT License[12] |
| Contextual-Top2Vec (Top2Vec) | BSD 3-Clause[13] |
| FASTopic / TopMost | Apache License 2.0[14] |
| ETM (reference implementation) | MIT License[15] |
| CTM (Contextualized Topic Models) | MIT License[16] |
| *Text datasets* | |
| OCTIS preprocessed corpora (20NG, BBC, M10, DBLP) | Original source terms via OCTIS dataset cards[17] |
| TrumpTweets (The Trump Archive) | Site terms / Twitter TOS[18] |
| *Biological datasets* | |
| PBMC3k (10x Genomics) | CC BY 4.0[19] |
| Zeisel mouse brain (GSE60361, GEO) | GEO usage policy (no explicit license)[20] |
| BreastCancer (Zenodo: 10.5281/zenodo.10620607) | CC BY 4.0[21] |

# D Additional Experiments

## D.1 More Qualitative Findings

To complement the qualitative findings presented in the main paper, we include additional quantitative analysis for BBC and M10 dataset here.

**BBC Dataset.** Figure 7a (MALLET) and Figure 8b (PRISM) display the top words for each topic on the BBC dataset with 5 topics. While MALLET produces reasonable topics, it redundantly captures politics in two separate themes and fails to isolate the entertainment domain. In contrast, PRISM yields distinct and semantically meaningful topics, effectively covering all major themes in the corpus. These results suggest that while MALLET offers a solid inference framework, our initialization method pushes the model further

---

[9]`https://github.com/MIND-Lab/OCTIS/blob/master/LICENSE`

[10]`https://github.com/RaRe-Technologies/gensim/blob/develop/LICENSE`

[11]`http://mallet.cs.umass.edu/` (see `LICENSE` in the distribution)

[12]`https://github.com/MaartenGr/BERTopic/blob/master/LICENSE`

[13]`https://github.com/ddangelov/Top2Vec/blob/master/LICENSE`

[14]`https://pypi.org/project/fastopic/` (license field)

[15]`https://github.com/adjidieng/ETM/blob/master/LICENSE`

[16]`https://github.com/MilaNLProc/contextualized-topic-models/blob/master/LICENSE`

[17]`https://github.com/MIND-Lab/OCTIS`

[18]`https://www.thetrumparchive.com/faq`; `https://developer.x.com/en/developer-terms`

[19]`https://www.10xgenomics.com/resources/datasets`

[20]`https://www.ncbi.nlm.nih.gov/geo/info/disclaimer.html`

[21]`https://zenodo.org/records/10620607`

| Topic 1 | Topic 2 | Topic 3 | Topic 4 | Topic 5 |
|---|---|---|---|---|
| company | technology | law | win | election |
| market | phone | case | good | government |
| rise | mobile | court | play | party |
| sale | game | government | game | labour |
| firm | service | rule | film | plan |
| price | music | claim | award | tory |
| share | user | legal | player | public |
| growth | computer | charge | back | country |
| economy | net | police | show | work |
| month | firm | ban | world | minister |

(a) MALLET Top 10 words

| Topic 1 | Topic 2 | Topic 3 | Topic 4 | Topic 5 |
|---|---|---|---|---|
| government | film | company | win | technology |
| election | good | market | game | computer |
| party | award | firm | play | phone |
| labour | music | rise | player | mobile |
| plan | win | sale | good | service |
| tory | show | price | back | user |
| law | include | economy | match | game |
| issue | star | share | team | firm |
| public | top | growth | club | net |
| minister | actor | month | final | music |

(b) PRISM Top 10 words

Figure 7: Top-10 words per topic on the **BBC** dataset with $K = 5$. Each column is a distinct topic. Colors in the panels denote manually interpreted categories ( politics, **entertainment**, **business**, sports, and **technology**); lighter shades indicate weaker relevance and white indicates no clear association. Panels: (a) MALLET, (b) PRISM.

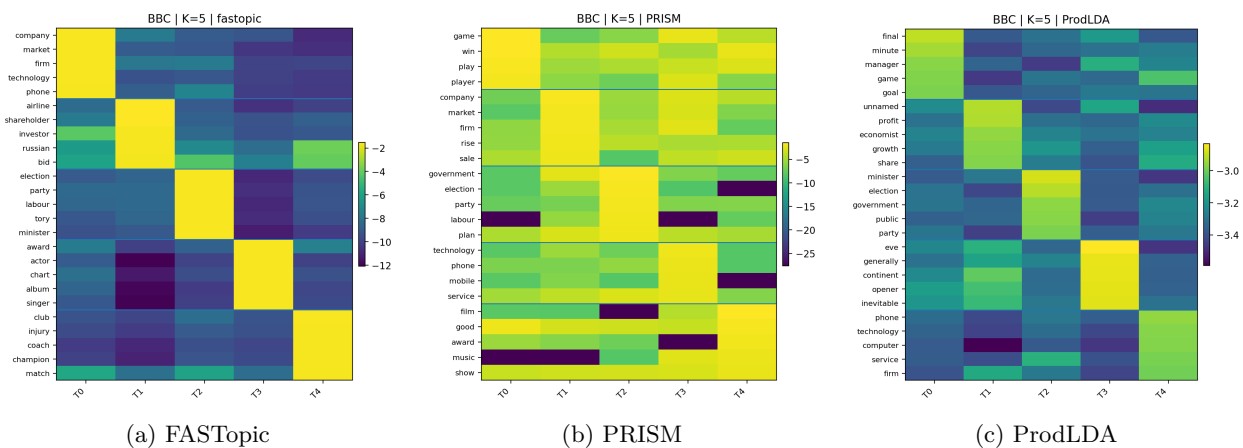

(a) FASTopic  (b) PRISM  (c) ProdLDA

Figure 8: Heatmaps on the **BBC** dataset with $K = 5$. Each column corresponds to a topic. Each column corresponds to a topic and each row to a top-ranked word (from the union of topics). Cell colors encode the topic-word relevance (i.e., $\phi_{k,w}$, shown on a log scale in the colorbar): lighter/yellow shades indicate higher probability mass, whereas darker/purple shades indicate lower probability. Panels: (a) FASTopic, (b) PRISM, (c) ProdLDA.

toward more coherent and semantically distinct topics, indicating that the observed improvements stem from our approach rather than the base model alone.

Figure 8 further corroborates these observations using topic–word relevance heatmaps. Although FASTopic and ProdLDA identify differentiated themes, some topics remain less cleanly interpretable: FASTopic's *T0* blends business and technology terms, and ProdLDA fails to recover a coherent entertainment topic (its *T3* is dominated by largely generic words), despite producing reasonable sports, politics, and technology-related themes elsewhere. In contrast, PRISM yields five clearly delineated topics aligned with the known BBC domains, concentrating high probability mass on consistent, domain-specific vocabularies while still allowing limited, context-driven lexical overlap across related topics. This graded overlap further motivates our use of PRISM for scRNA-seq, where genes can participate in multiple biological processes and thus benefit from representations that accommodate overlapping gene programs.

**M10 Dataset.** Figure 9 highlights contrasting qualitative behaviors on the heterogeneous M10 corpus. BERTopic yields highly coherent, easily nameable topics with an almost block-diagonal structure, indicating

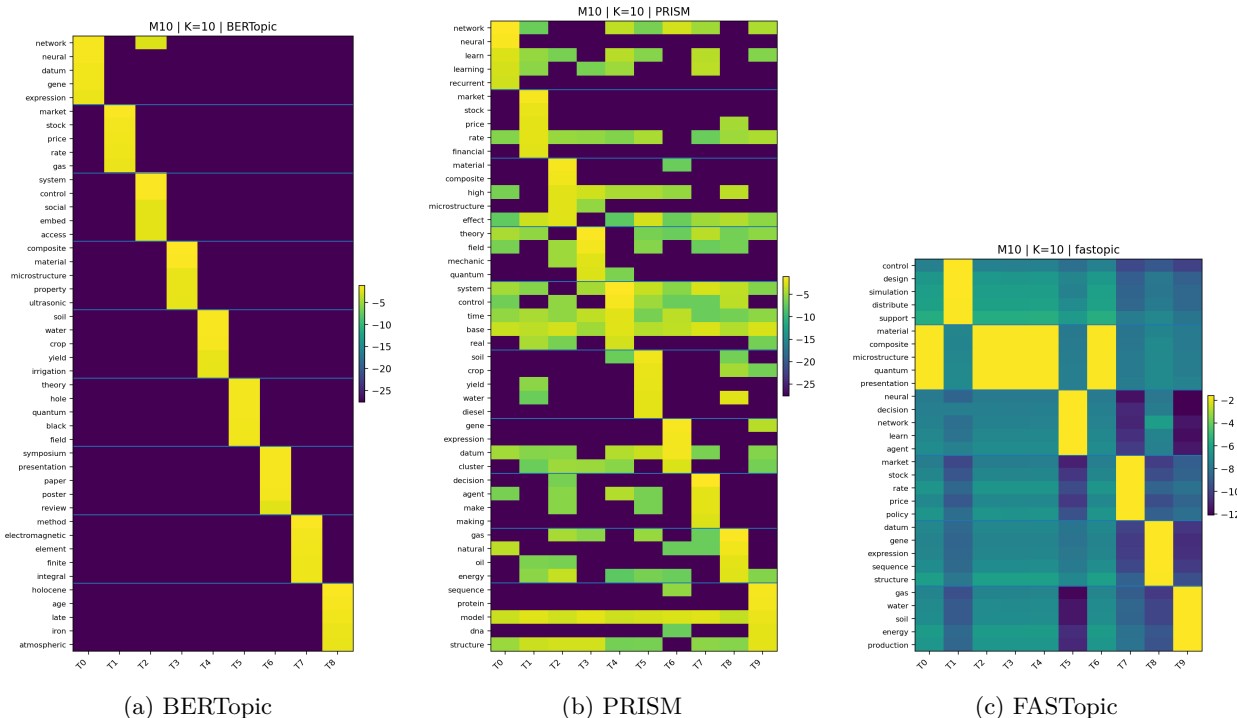

(a) BERTopic          (b) PRISM          (c) FASTopic

Figure 9: Heatmaps on the **M10** dataset with $K = 10$. Each column corresponds to a topic and each row to a top-ranked word (from the union of topics). Cell colors encode topic–word relevance (i.e., $\phi_{k,w}$, shown on a log scale in the colorbar): lighter/yellow shades indicate higher probability mass, whereas darker/purple shades indicate lower probability. Panels: (a) BERTopic, (b) PRISM, (c) FASTopic.

near-disjoint topic assignments and minimal redundancy; however, this sharp separation largely suppresses meaningful lexical sharing across related scientific contexts. FASTopic recovers several coherent themes (e.g., finance, learning, energy, and genomics), yet exhibits noticeable redundancy and blurred boundaries: materials/methods-related terms (e.g., *material*, *composite*, *microstructure*) receive elevated relevance across multiple topics, suggesting topic duplication rather than distinct semantic cores. In contrast, PRISM maintains clear topic centers while allowing selective, semantically plausible overlap; for example, while *gene* is most strongly associated with the gene-expression topic (*T6*), it retains non-negligible probability mass in the protein/structure topic (*T9*), reflecting context-dependent usage and enabling richer representations than either BERTopic's near-exclusive clusters or FASTopic's diffuse redundancy.

## D.2 Ablation: Priors and Embeddings

To assess the contribution of individual components of our method, we conducted ablation studies focusing on priors and embeddings. Specifically, we evaluated four variants: (1) replacing our custom word embeddings (constructed via soc-PPMI and diffusion maps) with pretrained GloVe embeddings; (2) introducing a random prior within the MALLET framework to test whether improvements arise solely from enabling an asymmetric $\boldsymbol{\beta}$ prior (unsupported in the original MALLET, which assumes a scalar $\boldsymbol{\beta}$); (3) using the raw PPMI matrix without applying cosine similarity, to test whether the latter indeed enhances the capture of indirect semantic associations beyond direct co-occurrence; and (4) substituting diffusion maps with SVD to examine whether SVD alone provides more effective embeddings. For each variant, we report both standard statistical metrics ($c_v$, NPMI) in Table 6 and WID task performance in Table 7. Overall, the ablations consistently underperformed the full model, confirming that both the embedding construction and the asymmetric $\boldsymbol{\beta}$ prior are necessary to achieve the observed gains.

### D.2.1 Statistic Metrics

We report ablation results using two standard coherence metrics, $c_v$ and NPMI, consistent with the evaluation used in the main paper.

Table 6: Coherence across datasets for ablation variants. Each entry is mean (std). Best is **bold**, second-best is underlined.

| MODELS | 20NewsGroup | | BBC | | M10 | | DBLP | | TrumpTweets | |
|---|---|---|---|---|---|---|---|---|---|---|
| | $c_v$ | NPMI | $c_v$ | NPMI | $c_v$ | NPMI | $c_v$ | NPMI | $c_v$ | NPMI |
| GloVe + GMM Prior | .6324 (.0021) | .1133 (.0076) | .6395 (.0119) | .1100 (.0099) | .4839 (.0129) | .0658 (.0169) | .4503 (.0310) | .0588 (.0177) | .4968 (.0085) | .0699 (.0091) |
| Random Prior | .6230 (.0053) | .1104 (.0104) | .6420 (.0096) | .1131 (.0137) | .4822 (.0484) | .0717 (.0332) | .4539 (.0476) | .0591 (.0214) | .4910 (.0078) | .0678 (.0028) |
| No SOC (PPMI→DM) | .6482 (.0103) | .1164 (.0058) | .6540 (.0219) | .1250 (.0072) | .5132 (.0018) | .0734 (.0231) | **.4879** (.0209) | **.0742** (.0114) | .5194 (.0037) | .0840 (.0079) |
| SVD in place of DM | .6436 (.0117) | .1159 (.0051) | .6456 (.0141) | .1115 (.0128) | .4814 (.0115) | .0694 (.0129) | .4611 (.0359) | .0616 (.0169) | .4969 (.0169) | .0697 (.0072) |
| **PRISM (reference)** | **.6592** (.0081) | **.1168** (.0061) | **.6781** (.0161) | **.1468** (.0179) | **.5285** (.0034) | **.0822** (.0132) | .4643 (.0152) | .0751 (.0131) | **.5571** (.0036) | **.0991** (.0112) |

The results in Table 6 highlight the consistent advantage of our method. Across all datasets, PRISM achieves the highest coherence scores, with the sole exception of DBLP, where it ranks second. Notably, the variant that omits cosine similarity (No SOC) emerges as the most competitive ablation, frequently attaining the second-best results. This suggests that the overall framework of our method effectively captures meaningful similarities, while the cosine-similarity step further enhances performance.

### D.2.2 Word Intrusion Detection

We also evaluate the ablation variants on the Word Intrusion Detection (WID) task, which was used in the main paper to assess topic interpretability. Results are reported in Table 7. Across all datasets, PRISM consistently outperforms the ablation baselines, achieving the highest WID accuracy. These findings indicate that PRISM produces more coherent and interpretable topics, enabling intruder words to be more reliably identified.

Table 7: Word Intrusion Detection (WID) accuracy for ablations across datasets. Each entry is mean accuracy (std) over 10 runs. Ablations: **GloVe Embed** (GloVe term embeddings → GMM; MoM to Dirichlet prior), **Random Prior** (random vector prior), **No SOC** (PPMI → DM; cosine-similarity step removed), **SVD in place of DM** (SOC-PPMI → SVD → GMM).

| MODELS | 20NewsGroup | BBC | M10 | DBLP | TrumpTweets |
|---|---|---|---|---|---|
| GloVe Embed | .5590 (.0480) | .5778 (.0072) | .3672 (.0492) | .2844 (.0376) | .2733 (.0618) |
| Random Prior | .5277 (.0494) | .5467 (.0091) | .3161 (.0547) | .2506 (.0448) | .2328 (.0627) |
| No SOC (PPMI→DM) | .5637 (.0492) | .5233 (.0145) | .3633 (.0577) | .2750 (.0467) | .2744 (.0687) |
| SVD in place of DM | .5543 (.0514) | .5533 (.0132) | .3589 (.0615) | .3133 (.0455) | .2411 (.0714) |
| PRISM (reference) | **.6099** (.0124) | **.6201** (.0281) | **.3921** (.0176) | **.3408** (.0512) | **.3057** (.0131) |

## D.3 Ablation: PPMI Context Window

### D.3.1 Statistic Metrics

We report ablation results using two standard coherence metrics, $c_v$ and NPMI, consistent with the evaluation used in the main paper. The results in Table 8 highlight the consistent advantage of our method. Across all datasets, PRISM achieves the highest coherence scores, whether documents are very short or long.

Table 8 compares PRISM priors constructed from PPMI estimated with fixed sliding windows ($w \in 2, 5, 10$) versus a document-level window. The results show a clear and consistent pattern: enlarging the context window generally improves coherence, but the document-level context yields the strongest and most stable coherence across all datasets. This is particularly important given the wide variation in average document length.

Table 8: Ablation on PPMI context window size. Each entry is mean (std) over three topic settings. Best is **bold**, second-best is underlined.

| MODELS | 20NG | | BBC News | | M10 | | DBLP | | TrumpTweets | |
|---|---|---|---|---|---|---|---|---|---|---|
| | $c_v$ | NPMI | $c_v$ | NPMI | $c_v$ | NPMI | $c_v$ | NPMI | $c_v$ | NPMI |
| PRISM (w=2) | .6472 (.0065) | .1030 (.0048) | .6459 (.0120) | .1250 (.0130) | .4482 (.0060) | .0645 (.0070) | .4460 (.0140) | .0580 (.0100) | .5250 (.0100) | .0780 (.0090) |
| PRISM (w=5) | .6499 (.0072) | .1108 (.0055) | .6534 (.0135) | .1202 (.0160) | .5035 (.0053) | .0765 (.0108) | .4575 (.0151) | .0710 (.0120) | .5310 (.0080) | .0910 (.0100) |
| PRISM (w=10) | .6538 (.0078) | .1115 (.0060) | .6592 (.0150) | .1319 (.0175) | .5046 (.0040) | **.0830** (.0120) | .4534 (.0150) | .0735 (.0130) | .5329 (.0160) | .0975 (.0110) |
| MALLET | .6322 (.0032) | .1018 (.0037) | .6384 (.0051) | .1285 (.0101) | .4571 (.0058) | .0671 (.0032) | .4329 (.0133) | .0467 (.0083) | .4989 (.0152) | .0701 (.0049) |
| **PRISM (doc-level)** | **.6592** (.0081) | **.1168** (.0061) | **.6781** (.0161) | **.1468** (.0179) | **.5285** (.0034) | .0822 (.0132) | **.4643** (.0152) | **.0751** (.0131) | **.5571** (.0036) | **.0991** (.0112) |

Table 9: Ablation on PPMI context window size for Word Intrusion Detection (WID) accuracy (mean (std) over three topic settings). Best is **bold**, second-best is underlined.

| MODELS | 20NewsGroup | BBC | M10 | DBLP | TrumpTweets |
|---|---|---|---|---|---|
| PRISM (w=2) | .5850 (.0180) | .5600 (.0300) | .3780 (.0220) | .3180 (.0500) | .2870 (.0180) |
| PRISM (w=5) | .5975 (.0150) | .5950 (.0260) | .3860 (.0190) | .3310 (.0480) | .2980 (.0150) |
| PRISM (w=10) | .6068 (.0130) | .6150 (.0240) | .3910 (.0180) | .3385 (.0520) | .2996 (.0104) |
| MALLET | .5681 (.0206) | .4689 (.0412) | .3711 (.0406) | .3031 (.0554) | .2697 (.0302) |
| **PRISM (doc-level)** | **.6099**[†](.0124) | **.6201**[†](.0281) | **.3921**[†](.0176) | **.3408**[†](.0512) | **.3057**[†](.0131) |

### D.3.2 Word Intrusion Detection

We also evaluate the ablation variants on the Word Intrusion Detection (WID) task, which was used in the main paper to assess topic interpretability. Results in Table 9 mirror the coherence trends and strengthen the justification for our default choice: PRISM with a document-level context window attains the highest WID accuracy across all datasets (and consistently exceeds the fixed-window variants). Because WID measures whether an intruder can be identified from a topic's top words, higher accuracy indicates that the topic word lists are more coherent to humans.

### D.4 Complexity Analysis

Table 10: Dominant time and space complexities.

| Model | Time (dominant) | Space (dominant) |
|---|---|---|
| *BoW topic/factor models* | | |
| LDA | $O(\text{nnz}(X)\,K) + O(|V|K)$ | $O(\text{nnz}(X) + NK + |V|K)$ |
| NMF | $O(\text{nnz}(X)\,K)$ | $O(\text{nnz}(X) + NK + |V|K)$ |
| ProdLDA | $O(\text{nnz}(X)\,K) + O(|\Theta|)$ | $O(\text{nnz}(X) + NK + |V|K + |\Theta|)$ |
| ETM | $O(\text{nnz}(X)\,K) + O(|V|K)$ | $O(\text{nnz}(X) + NK + |V|K)$ |
| CTM | $O(\text{nnz}(X)\,K) + O(|\Theta|)$ | $O(\text{nnz}(X) + NK + |V|K + |\Theta|)$ |
| MALLET | $O(\text{nnz}(X)\,\bar{K}_{\text{act}})$ | $O(\text{nnz}(X) + NK + |V|K)$ |
| *Embedding → clustering pipelines* | | |
| BERTopic | $O(N\,c_{\text{enc}}) + O(N\log N) + O(N^2) + O(|V|K)$ | $O(N\,d_{\text{emb}} + |V|K)$ |
| Contextual-Top2Vec | $O(N\,c_{\text{enc}}) + O(N\log N) + O(N^2)$ | $O(N\,d_{\text{emb}})$ |
| FASTopic | $O(\text{nnz}(X)) + O(|V|K^2) + O(|V|K)$ | $O(\text{nnz}(X) + |V|K)$ |
| *Our model* | | |
| PRISM (ours) | $O(\text{nnz}(X)\,\bar{K}_{\text{act}}) \; + \; O(|V|^3)$ | $O(\text{nnz}(X) + NK + |V|K) + O(|V|^2)$ |

**Reading Table 10.** *Symbols.* $N$: #documents; $|V|$: vocabulary size; $K$: #topics; $X \in \mathbb{R}^{N \times |V|}$ sparse with nnz$(X)$ nonzeros; $|\Theta|$: number of trainable neural parameters (encoder/decoder); $c_{\text{enc}}$: per-document embedding cost of the text encoder; $d_{\text{emb}}$: embedding dimension; $\bar{K}_{\text{act}}$: average number of *active* topics per

Table 11: Wall-clock runtime in seconds (mean (std) over three runs).

| Model | 20NewsGroup | BBC | M10 | DBLP | TrumpTweets |
|---|---|---|---|---|---|
| **MALLET** | 38.24 (6.81) | 32.65 (0.58) | 31.11 (0.23) | 32.26 (1.69) | 31.07 (0.61) |
| **PRISM** | 46.67 (12.42) | 38.08 (2.10) | 39.56 (5.29) | 42.67 (4.04) | 34.67 (2.52) |
| **BERTopic** | 668.14 (7.50) | 130.06 (5.46) | 102.30 (21.63) | 2005.01 (51.50) | 625.45 (42.38) |

token under SparseLDA (typically $\bar{K}_{\mathrm{act}} \ll K$). Time entries are per iteration/epoch for iterative models and per pipeline for embedding→clustering methods; space lists dominant stored objects.

**BoW models.** LDA: $O(\mathrm{nnz}(X)\,K)$ from local doc–topic variational/sufficient–statistic updates (each nonzero token touches $K$ topic scores); $O(|V|K)$ from global topic–word updates/normalization; space $O(\mathrm{nnz}(X) + NK + |V|K)$ for sparse data plus doc–topic and topic–word matrices. NMF: $O(\mathrm{nnz}(X)\,K)$ from multiplicative updates touching nonzeros per factor; same space pattern as BoW factorization. ProdLDA: $O(\mathrm{nnz}(X)\,K)$ (local BoW term) $+ O(|\Theta|)$ from backprop through amortized encoder/decoder per epoch; space adds $|\Theta|$. ETM: $O(\mathrm{nnz}(X)\,K)$ (local inference) $+ O(|V|K)$ for topic–embedding softmax/global updates; BoW space. CTM: $O(\mathrm{nnz}(X)\,K)$ (local) $+ O(|\Theta|)$ (backprop); BoW space $+ |\Theta|$. MALLET: $O(\mathrm{nnz}(X)\,\bar{K}_{\mathrm{act}})$ since per-token sampling operates only over nonzero counts for the (few) topics active in that context; space as BoW.

**Embedding→clustering.** BERTopic: $O(N\,c_{\mathrm{enc}})$ for encoder forward passes; $O(N \log N)$ from ANN/kNN graph (UMAP build); $O(N^2)$ worst-case from HDBSCAN; $O(|V|K)$ from c-TF-IDF labeling; space $O(N\,d_{\mathrm{emb}})$ for embeddings $+ O(|V|K)$ for cluster-term weights. Contextual-Top2Vec: same $O(N\,c_{\mathrm{enc}}) + O(N \log N) + O(N^2)$ (encoder+UMAP+HDBSCAN); no c-TF-IDF term; space $O(N\,d_{\mathrm{emb}})$. FASTopic: $O(\mathrm{nnz}(X))$ to build word co-occurrences; $O(|V|K^2)$ for anchor selection; $O(|V|K)$ for recovery; space $O(\mathrm{nnz}(X) + |V|K)$.

**PRISM (ours).** Inference matches MALLET: $O(\mathrm{nnz}(X)\,\bar{K}_{\mathrm{act}})$ time; $O(\mathrm{nnz}(X) + NK + |V|K)$ space. The one-time word-embedding pre-step (dense case) is dominated by the diffusion-maps spectral decomposition on a $|V| \times |V|$ kernel: $\approx O(|V|^3)$ time and $O(|V|^2)$ space (the PPMI transform and all-pairs similarities are subsumed).

**Reading Table 11.** As shown in Table 11, PRISM exhibits runtimes close to MALLET across all datasets (34.67-46.67s vs. 31.07-38.24s), indicating that the corpus-intrinsic prior construction and initialization introduce only modest additional overhead. In contrast, BERTopic is substantially slower (102.30-2005.01s), consistent with the cost of contextual embedding extraction and subsequent dimensionality reduction and clustering, whose runtime scales strongly with the number of documents (e.g., DBLP). Overall, these results suggest that PRISM largely preserves the computational profile of classical LDA-style training while delivering the empirical improvements reported elsewhere in the paper.

## E    Biological Experiments

### E.1    Biological Technical Details

**Bio Preprocessing.** We first filtered lowly detected genes, retaining only those expressed in at least $m$ cells using `scanpy.pp.filter_genes` (Wolf et al., 2018). Counts were normalized per cell to a fixed library size (`scanpy.pp.normalize_total`) and log-transformed (`scanpy.pp.log1p`) (Wolf et al., 2018; Luecken & Theis, 2019). Highly variable genes (HVGs) were then selected with the Seurat v3 method (`flavor=seurat_v3`), using raw counts via `layer="counts"` and, where applicable, merging HVGs across batches via `batch_key` (scverse, accessed 2025; Stuart et al., 2019). Analyses were performed in Scanpy with AnnData as the primary container (Wolf et al., 2018; Virshup et al., 2024).

**PRISM adaptation for scRNA-seq.** Our workflow matches the text-corpus setup except for the PPMI construction. In text, the sliding window is an entire document; in scRNA-seq, we first construct a cell–cell

$k$-NN (here $k = 30$) on PCA features (Euclidean distance), a standard step in single-cell pipelines, and treat each cell's neighborhood as the PPMI 'window'. We then compute gene–gene PPMI from co-occurrences across these neighborhoods. The intuition is that genes co-expressed across similar cells (often sharing cell type/state) better capture shared programs than mere co-occurrence within a single cell, where multiple unrelated programs may be intermixed.

### E.2 Biological Evaluations

### E.2.1 Gene-Set Quality Metrics

We assess the biological plausibility of inferred gene expression programs using three complementary criteria: (i) enrichment—overrepresentation of curated pathways/functions (e.g., GO/KEGG/MSigDB) relative to a fixed gene universe; (ii) coherence—within–gene-set co-variation of expression across cells; and (iii) coverage–the extent to which programs collectively recover pathway members. Metrics are computed per program and then aggregated per model and dataset. We control the false discovery rate (FDR)–the expected proportion of false positives among rejected hypotheses–using the Benjamini–Hochberg procedure and report BH-adjusted $q$-values (Benjamini & Hochberg, 1995).

**Enrichment strength $(-\log_{10} q)$.** Let $U$ be the gene universe $(|U| = N)$. For a program $S \subseteq U$ $(|S| = K)$ and a curated pathway $P \subseteq U$ $(|P| = M)$, with overlap $x = |S \cap P|$, we test overrepresentation using a one-sided hypergeometric (Fisher's exact) test:

$$p = \sum_{i=x}^{\min(K,M)} \frac{\binom{M}{i}\binom{N-M}{K-i}}{\binom{N}{K}}.$$

We then correct the collection of $p$-values across all tested pathways for $S$ using FDR (Benjamini–Hochberg) to obtain $q$, and report the strength as $-\log_{10}(q)$ (higher is better).

**Coherence (mean within-set Spearman correlation).** Given the cells×genes expression matrix, compute Spearman correlations $\rho_{ij}$ across cells for all gene pairs $i < j$, $i, j \in S$, and average:

$$\text{coherence} = \frac{2}{K(K-1)} \sum_{i<j, \ i,j\in S} \rho_{ij}.$$

Higher values indicate coordinated expression consistent with a functional program.

**Coverage (pathway completeness).** Per-pathway coverage is the recovered fraction of members:

$$\text{cov}(P \mid S) = \frac{|S \cap P|}{|P|}.$$

Aggregate over the set $\mathcal{P}^*$ of pathways significantly enriched for $S$ (post-FDR), e.g., by a simple mean:

$$\text{coverage} = \frac{1}{|\mathcal{P}^*|} \sum_{P \in \mathcal{P}^*} \text{cov}(P \mid S),$$

or a size-weighted mean (specified in the Supplement). Higher is better.

### E.2.2 LLM Confidence Score

To assess the biological relevance of gene sets derived from topic models, we follow the LLM-based evaluation protocol introduced by (Hu et al., 2025).[22] The core idea is to query a large language model (GPT-4) with each gene set and evaluate whether it can (1) identify a coherent biological process (BP) associated with the gene set, and (2) express high confidence in that association.

---

[22] https://www.ncbi.nlm.nih.gov/pmc/articles/PMC11725441

**Prompt Design.** Each gene set is presented in a natural language prompt, instructing the model to infer a shared biological process based on the listed genes. Prompts are carefully crafted to be neutral and avoid leading the model toward specific functions. The model is then asked to (i) name the most likely BP, and (ii) rate its confidence on a scale from 0 to 1.

**Scoring.** The model's textual output is manually inspected to verify whether the inferred BP matches a plausible biological function supported by external evidence (e.g., GO annotations). Confidence scores are recorded for each gene set and aggregated to assess overall coherence across topics.

**Interpretation.** As shown in prior work, gene sets yielding low confidence often correspond to functionally inconsistent or noisy groups, whereas high-confidence predictions align with known biological pathways. Thus, the LLM confidence score serves as a proxy for functional coherence and interpretability of the gene sets.

### E.3 Biological Quality Results

Table 12: LLM-based evaluation scores (mean and std over 10 runs) across biological datasets.

| Models | BreastCancer | pbmc3k | zeisel_brain |
|---|---|---|---|
| cNMF | .8000 (.0505) | **.9021** (.0401) | .9330 (.0172) |
| scHPF | .7712 (.0240) | .8326 (.0235) | .8980 (.0658) |
| NMF | .7880 (.0331) | .6590 (.0141) | .6670 (.0435) |
| ProdLDA | .7600 (.0469) | .4950 (.1862) | .5950 (.0282) |
| MALLET | .7920 (.0352) | .8392 (.0276) | .8920 (.0268) |
| PRISM | **.8320**[†](.0171) | .8997[†](.0167) | **.9350**[†](.0206) |

As shown in Table 12, PRISM attains the highest GPT-4 confidence on two of three datasets (BreastCancer, zeisel_brain) and is a close second on pbmc3k, consistently outperforming MALLET. Together with the gene-set metrics (see Table 3), these results may indicate that corpus-intrinsic LDA variants are well-suited to scRNA-seq: PRISM and MALLET clearly surpass generic text models (NMF, ProdLDA), while PRISM is competitive with single-cell baselines (cNMF, scHPF) and reliably improves over MALLET. Since GPT-4 scores track biological plausibility (Hu et al., 2025), PRISM's gains suggest more coherent, biologically meaningful topics.

