# OpenReview forum: "PRISM: PRIor from corpus Statistics for topic Modeling"
_TMLR — Accepted by TMLR_

### Review · Reviewer_xMSF · 2025-12-14

**Summary Of Contributions:**

This paper introduces PRISM, a novel corpus-intrinsic initialization method for Latent Dirichlet Allocation. The core contribution lies in deriving a data-driven Dirichlet prior parameter from word co-occurrence statistics within the corpus itself, effectively initializing LDA without relying on external embeddings or pre-trained models. This is particularly valuable in resource-constrained settings and emerging domains where external knowledge may be limited or unreliable. The authors demonstrate improved topic coherence and interpretability across diverse datasets, rivaling the performance of embedding-based methods. The method appears robust and offers a practical alternative for enhancing topic modeling in scenarios where external knowledge is scarce.

**Additional Comments:**

Overall, the paper is clearly written and tackles a relevant problem in topic modeling. The corpus-intrinsic perspective is a reasonable design choice, and the empirical evaluation spans a diverse set of datasets. However, several aspects limit the current impact of the work. In particular, the methodological contribution appears largely incremental relative to prior literature, key design choices lack sufficient ablation and justification, and the evaluation relies on proxies that raise concerns about interpretability and robustness. Addressing these issues, along with the other requested revisions, would substantially improve the technical rigor and clarify the paper’s true contribution to the field.

**Audience:**

Yes

**Audience Explanation:**

The TMLR audience includes researchers and practitioners working in areas such as NLP, ML, Bayesian methods, and bioinformatics.  Topic modeling is a fundamental technique in these fields, and the problem of improving topic coherence and interpretability remains an active area of research. PRISM offers a novel solution that addresses the limitations of existing methods, particularly in resource-constrained settings. The application to single-cell RNA sequencing data is also of significant interest to the bioinformatics community.  The corpus-intrinsic approach and demonstrated performance make this paper relevant and valuable to a broad segment of the TMLR readership.

**Broader Impact Concerns:**

The ethical implications of this work are relatively minor. Topic modeling itself can be used for various purposes, some of which may have ethical concerns (e.g., biased topic discovery). However, PRISM does not introduce any new ethical risks beyond those associated with standard topic modeling techniques.

**Claims And Evidence:**

Yes

**Claims Explanation:**

The provided experiments consistently report improvements in topic coherence and interpretability when using PRISM compared to other LDA baselines. The performance is also shown to be competitive with embedding-based methods. The authors explicitly state that the improvements are demonstrated across five diverse datasets, suggesting generalizability.  The comparison to models like MALLET, NMF, ProdLDA, cNMF, and scHPF provides a reasonable benchmark. The claim of robustness is supported by consistent performance across different datasets and domains.

However, the reliance on GPT-4–generated scores to assess biological plausibility in the scRNA-seq experiments is problematic. For instance, the cited paper itself notes that while the generated analysis text was largely factual, GPT-4 occasionally produced unverifiable statements, underscoring that even state-of-the-art LLMs require explicit fact-checking and reference validation, either automated or manual. These scores lack transparency, reproducibility, and a clear biological grounding, making it difficult to evaluate what aspects of the data they actually validate. Without a principled justification or comparison to established domain-specific metrics or expert annotations, the use of a general-purpose language model risks introducing opaque biases rather than providing meaningful biological credibility.

**Requested Changes:**

* **[major]** Provide a more thorough ablation study evaluating PRISM’s sensitivity to key hyperparameters in the core methodology, such as those governing the PPMI computation. This is necessary to assess robustness and to provide actionable guidance for practical use.

* **[major]** The contribution appears largely incremental. The proposed method heavily builds upon existing topic modeling and initialization techniques, reusing well-established components such as PPMI-based co-occurrence statistics, diffusion-based embeddings, and standard clustering pipelines. While the integration is technically sound, the paper offers limited conceptual novelty beyond combining known techniques, and the empirical gains over prior work are relatively modest. This raises questions about whether the method represents a substantive advance over existing approaches or primarily a refinement of prior art.

* **[major]** While GPT-4–based scores are used as a proxy for biological plausibility in the scRNA-seq experiments, the evaluation would be strengthened by incorporating additional quantitative metrics. Notably, the cited work itself acknowledges that although GPT-4–generated analyses are largely factual, the model occasionally produces unverifiable statements, highlighting the necessity of explicit fact-checking and reference validation, whether automated or manual.

* **[minor]** Provide a clearer and more detailed description of the diffusion maps algorithm used to construct embeddings, including sufficient detail to ensure reproducibility.

* **[minor]** Seems like some parts of the Figure 1 are hand-drawn. It would be good to have them in a vector format with proper fonts.

---

> ### Author Response · Authors · 2026-02-25
>
> **1.** Thank you for requesting a more thorough sensitivity analysis. We added a dedicated ablation study on the PPMI context-window size (Tables 8-9 in Appendix D.3), reporting results across multiple datasets and topic counts for coherence ($c_v$, NPMI) and Word Intrusion Detection (WID). Across $w \in \{2,5,10\}$ we observe stable trends and limited sensitivity, with window sizes between 5 and 10 consistently performing best. We also report comparisons to MALLET and to the document-level variant to contextualize this effect. Due to space constraints, the full ablation appears in the Appendix.
>
> **2.** We appreciate this framing request and clarified the contribution accordingly. PRISM’s novelty is not any single ingredient (PPMI, diffusion maps, clustering), but their integration to reshape LDA’s inductive bias. While existing work typically optimizes low-dimensional concentration parameters ($\boldsymbol{\alpha}, \boldsymbol{\beta}$) to regulate global sparsity (Burkhardt & Kramer, 2019; Joo et al., 2020), PRISM produces a $V$-dimensional Dirichlet prior whose structure is derived entirely from corpus-internal signals. This enables vocabulary-level semantic structure to be injected into classical LDA without relying on external embeddings or replacing the generative framework. Empirically, PRISM consistently narrows the gap between MALLET-based LDA and SOTA approaches that depend on large-scale external pretraining (e.g., BERTopic, FASTopic), suggesting that substantial gains attributed to external representations can be recovered via structured corpus-intrinsic priors, particularly relevant when pretrained semantics are misaligned or unavailable.
>
> **3.** Thank you for the important caution regarding GPT-based analysis. We agree that it should not be the primary quantitative evidence for biological plausibility. Our experiments already include statistical biological validation metrics (Table 12); to better reflect their role, we moved this table into the main paper (now Table 3) and repositioned the GPT-based interpretation to the Appendix as supplementary qualitative evidence (Table 12). This ensures that biological validation is grounded in explicit quantitative measures.
>
> **4.** We expanded Section 4.1.2 to fully specify the diffusion-maps pipeline, including construction of the transition matrix, density normalization, eigenvector selection, diffusion time, and the choice of embedding dimension.
>
> **5.** We adjusted Figure 1 following your advice.

---

> > ### Comment · Reviewer_xMSF · 2026-02-26
> >
> > Thank you for the revisions. The paper is in much better shape now. The main contributions are clearly stated and the overall presentation is more coherent.

---

### Review · Reviewer_bU34 · 2026-01-06

**Summary Of Contributions:**

This paper proposes PRISM, a corpus-intrinsic way to improve LDA without using any external knowledge. It initializes a topic–word Dirichlet prior by computing a second-order word similarity graph from PPMI along with diffusion map embeddings. Therefore, this study does not changes LDA’s generative process but improves topic representations. Empirically, across five text datasets and three single-cell RNA sequence datasets, PRISM consistently improves coherence and interpretability over baseline methods.

**Additional Comments:**

NA

**Audience:**

Yes

**Audience Explanation:**

The paper offers a corpus-intrinsic way to improve classical LDA via an informative Dirichlet prior, and demonstrates consistent gains on both text corpora and scRNA-seq. Therefore, the TMLR readers in probabilistic modeling and representation learning may be interested in this work.

**Claims And Evidence:**

Yes

**Claims Explanation:**

The proposed method is grounded in established technqiues such as PPMI and diffusion maps with a reasonable theoretical support.  The claims are supported by consistent improvements across multiple text and a scRNA-seq datasets, evaluated with quantitative metrics such as topic coherence and an LLM-assisted word-intrusion test.

**Requested Changes:**

1. Justify “unbiased pattern discovery” in Introduction by adding citations, evidence or explanations. It is not clear why corpus-intrinsic methods are better than methods using external knowledge.

2. Provide more contexts about "Second-order similarity" in Section 3.2.

3. Section 2 Related Work only lists the relevant works without discussing limitations.

4. Section 2.1 discusses topic model with expert knowledges but it does not cover recent works (e.g., within 5 years). It may include some works like [1,2].

[1] Automatic phenotyping by a seed-guided topic model. KDD, 2022.

[2] Modeling electronic health record data using an end-to-end knowledge-graph-informed topic model. Scientific Report, 2022.

5. Update related work in Section 2.2 (“Complementary work has examined …”). The discussion cites older references (latest around 2009). Please add a few more recent works (e.g., within 5 years).

6. Remove section 4.1 or merge it with the above section. It has been discussed in Introduction.

7. In Sec. 4.2.1, please clarify that W is a weighted similarity graph (i.e., a weighted adjacency matrix).

8. Qualitative analysis could add a heatmap to show some topics and their top words with probabilities. Moreover, the visualizations lack expert-knowledge–guided baseline such as SeededLDA.

---

> ### Author Response · Authors · 2026-02-25
>
> **1.** We appreciate this feedback and revised the Introduction to sharpen what we mean by “unbiased pattern discovery.” We now explain that pretrained representations may carry artifacts from their pretraining distributions and encode inductive biases that can be misaligned with specialized or data-scarce domains. We support this discussion with recent references and also cite an ablation (Appendix D.2) showing that injecting external embeddings (GloVe) reduces performance in our setting.
>
> **2.** Thank you for the suggestion. Section 3.2 has been expanded to provide additional intuition for second-order similarity. We include an illustrative example for intuition, and theoretical-empirical references linking PPMI-based similarity to distributional semantics and implicit matrix factorization.
>
> **3.** We revised Section 2 to move beyond listing prior work and instead articulate the assumptions and limitations of both externally guided and corpus-intrinsic approaches. In particular, we clarify that knowledge-guided and embedding-based methods depend on curated resources or pretrained semantics, which may be less reliable when modality mismatch and measurement noise weaken conceptual alignment (e.g., in single-cell settings) (Kedzierska et al., 2025). We also sharpen the contrast to modern Dirichlet hyperparameter learning (Burkhardt & Kramer, 2019; Joo et al., 2020), which primarily adjusts low-dimensional concentration parameters to control global sparsity, whereas PRISM introduces a $V$-dimensional, semantically structured prior derived entirely from corpus-intrinsic signals. These edits make PRISM’s positioning and scope more explicit.
>
> **4.** We incorporated your point by updating Section 2.1 with recent expert-knowledge--guided topic modeling papers (e.g., KDD 2022; Scientific Reports 2022), ensuring the related-work coverage reflects developments from the last five years.
>
> **5.** Following your suggestion, Section 2.2 now includes additional discussion of Dirichlet hyperparameter learning in neural topic models (e.g., Burkhardt & Kramer, 2019; Joo et al., 2020). We use these citations to highlight that many modern variants still treat $\alpha$ and $\beta$ largely as global concentration parameters, reinforcing the distinction from PRISM’s vocabulary-structured prior.
>
> **6.** To reduce redundancy, we merged Section 4.1 with the preceding section and edited for smoother flow.
>
> **7.** We clarified in Section 4.2.1 that $W$ denotes a weighted similarity graph (i.e., a weighted adjacency matrix).
>
> **8.** We added a qualitative heatmap visualization of representative topics and their top-word probabilities (Appendix D.1). Because of space limitations, we place the figure in the appendix. We also considered the suggested SeededLDA baseline  (Jagarlamudi et al., 2012). Seed-guided models require curated seed sets and therefore introduce semi-supervision and dataset-specific engineering; performance can also be sensitive to seed choice, complicating systematic comparison. For this reason we focus on fully unsupervised baselines. Notably, we already compare against strong externally informed topic models (BERTopic, FASTopic, Contextual-Top2Vec) that leverage large pretrained encoders. We agree that combining corpus-intrinsic priors with seed guidance is interesting and leave it as future work.

---

> > ### Comment · Reviewer_bU34 · 2026-02-25
> > **Update comments**
> >
> > I think this paper is well-written and currently has no issues.

---

### Review · Reviewer_AEEe · 2026-02-16

**Summary Of Contributions:**

The paper provides a way to initialize the word-topic embeddings by learning them from the corpus. The embedding method casts the words as nodes in a graph and extracts embeddings from the diffusion map of the graph. The motivation of developing corpus-intrinsic embeddings, and the method is interesting. The method is also presented in an intuitive manner. However, the writing itself requires work in terms of substantiating several claims present throughout the paper.

**Audience:**

Yes

**Audience Explanation:**

The idea is an interesting application of spectral methods to learning useful embeddings for LDA initialization.

**Broader Impact Concerns:**

I don't have any concerns.

**Claims And Evidence:**

No

**Claims Explanation:**

MANY of the sentences are implicit claims about the literature, but unsubstantiated. E.g., “foundation models are underdeveloped and existing knowledge is often fragmented” , “..pre-trained models…fall short in data-scare settings” etc.

This is is a major issue, and forces the reader to take these claims at face value, which is not ideal.

**Requested Changes:**

- The paper (especially the introduction) needs a fine-grained analysis of the implicit claims and whether they are supported by the literature. Otherwise, it is hard to convince the reader of the motivation of the paper.
- I could not tell what the value of “t” is in the experiments. More broadly, I think the paper will benefit from ablations/sensitivity analysis and discussion of the hyperparameters it introduces to the LDA pipeline (I appreciate the practical guidance for m).
- While I appreciate the complexity analysis, I think that as a practitioner, I still cannot develop a mental model around the sensitivity of the method. For example, how many docs are needed to build reliable embeddings? What is the interpretation of the embeddings themselves? - Related to the previous point, while the main contribution of the paper is around the word-embedding method, the paper is missing discussion around how good these embeddings are on their own. Technically, LDA is just a downstream application to this method, but the method is general. Is there a reason the paper couples them?
- For Section 6, why was K set to 5? And why were other baselines from Section 5 not used? Again, a justification/discussion would help the section.
- For my understanding: Why did you choose PPMI as the metric to define edge weights? What are other choices? Why not them?

---

> ### Author Response · Authors · 2026-02-25
>
> **1.** Thank you for emphasizing the need to better justify the motivational statements in the Introduction. We rewrote the motivation to be more granular and evidence-based. Specifically, we replaced broad claims about pretrained models with a tighter framing in terms of distribution shift and modality/representation misalignment, and we added supporting citations on pretraining-induced artifacts and misalignment (Belém et al., 2024; Ghate et al., 2025). We also make clear that our argument is not that externally informed methods are generally worse, but that corpus-intrinsic approaches can be preferable when curated knowledge or pretrained semantics are incomplete or not optimized for the target modality. We support this discussion with recent references and also cite an ablation (Appendix D.2) showing that injecting external embeddings (GloVe) reduces performance in our setting.
>
> **2.** We appreciate this clarification request. Section 4.1.2 (“From Graph to Embeddings”) now states the diffusion-time parameter explicitly, and we note that we fix $t=1$ in all experiments. We also added a short rationale: $t=1$ retains local neighborhoods and fine-grained structure, while larger $t$ increasingly attenuates high-frequency components and can over-smooth embeddings toward global structure. More generally, we expanded the reporting and discussion of PRISM hyperparameters, including density normalization ($\alpha=1$), diffusion time ($t=1$), and embedding dimension $m$, and we include additional ablations over key parameters (e.g., PPMI window size and $m$) in the Appendix D.3.
>
> **3.** Thank you for this thoughtful set of questions.
>   *Corpus-size sensitivity.* Our datasets span a range of corpus sizes (Appendix A), and PRISM improves over MALLET across this spectrum, suggesting it is not tied to a narrow data regime. A dedicated scaling study is valuable and we list it as future work.
>   *Interpreting and validating the embeddings.* The diffusion embeddings capture corpus-internal geometry: words with similar PPMI-based contextual neighborhoods lie nearby, and leading components reflect increasingly global structure. While we do not evaluate them on unrelated downstream tasks, we validate their utility through controlled substitutions and ablations. In particular, replacing PRISM embeddings with pretrained GloVe degrades performance (Appendix D.2), and additional ablations isolate the contribution of each construction step.
>   *Why couple to LDA.* Our choice is motivated by the biological domain: the LDA generative process (documents as mixtures of topics; topics as mixtures of words) is a well-recognized analog for single-cell gene expression (cells as mixtures of biological processes; processes as mixtures of genes). By coupling our method with LDA, we provide a model that is both semantically informed and remains highly interpretable for biologists, as we demonstrate in Section 6.
>
> **4.** Thanks for raising this point. In the scRNA-seq experiments we set $K=5$ to focus on coarse gene programs aligned with broad biological processes rather than fine substructure. Keeping $K$ fixed also supports controlled comparisons by holding model capacity constant and isolating the impact of initialization. We additionally verified that larger $K$ yields more fragmented gene sets, as expected, while preserving the main relative performance trends across methods. Regarding baselines, several methods in Section 5 rely on pretrained textual embeddings or contextual document representations that are not defined for gene-expression matrices. We therefore compare against modality-agnostic, corpus-intrinsic baselines (MALLET, ProdLDA, NMF). Notably, ProdLDA and NMF underperform on the biological dataset, consistent with the broader challenge of transferring architectures designed around text-embedding pipelines to scRNA-seq.
>
> **5.** Our goal is to build the word-word graph from corpus-internal statistics only. PMI quantifies association by contrasting co-occurrence with independence (Church & Hanks, 1990). Because raw PMI can be unstable for low-frequency events, we use PPMI to retain informative positive associations while suppressing noisy negative values. This choice is supported theoretically and empirically: Levy and Goldberg ("Neural Word Embedding as Implicit Matrix Factorization") show that SGNS implicitly factorizes a shifted PPMI matrix, connecting PPMI to widely used predictive embeddings, and prior work demonstrates that PPMI with cosine similarity is competitive for semantic similarity  (Bullinaria & Levy, 2012). Alternatives such as raw counts or conditional probabilities are more sensitive to frequency effects, whereas neural embeddings introduce additional modeling assumptions. PPMI provides a statistically grounded, and fully corpus-intrinsic representation aligned with our design goals.

---

> > ### Comment · Reviewer_AEEe · 2026-03-02
> >
> > Thank you for the addressing my comments, and I appreciate the clarifications / discussions included in the paper. I do not have any further concerns.

---

### Author Response · Authors · 2026-03-16

Dear Kejun,
We'd be thankful to kindly inquire about the status of our article, as all three reviewers acknowledged, after reading our author responses, that they do not have any further concerns regarding the article.

Thank you

---

### Decision · Action_Editor_U3m3 · 2026-03-19

**Recommendation:** Accept as is

**Audience:**

Yes

**Audience Explanation:**

Word embedding and topic modeling are classical subjects that are of interest to many people.

**Claims And Evidence:**

Yes

**Claims Explanation:**

This work provides a better way to initialize LDA, and experimental results support the claim.